# Exploiting Client Heterogeneity for Forgetting Mitigation in Federated Continual Learning: A Spatio-Temporal Gradient Alignment Approach

## Abstract

Federated Continual Learning (FCL) has recently emerged as a crucial research area, as data from distributed clients typically arrives as a stream, requiring sequential learning. This paper explores a more practical and challenging FCL setting, where clients may have unrelated or even heterogeneous tasks, leading to gradient conflicts where local updates point in divergent directions. In such a scenario, statistical heterogeneity and data noise can create spurious correlations, leading to biased feature learning and severe catastrophic forgetting. Existing FCL approaches often use generative replay to create pseudo-datasets of previous tasks. However, generative replay itself suffers from catastrophic forgetting and task divergence among clients, leading to overfitting phenomenon. To address these challenges, we propose a novel approach called **S**patio-**T**emporal gr**A**dient align**M**ent with **P**rototypical Coreset (STAMP). Our contributions are threefold: 1) We develop a model-agnostic method to determine subset of samples that effectively form prototypes when using a prototypical network, making it resilient to continual learning challenges; 2) We introduce a spatio-temporal gradient alignment approach, applied at both the client-side (temporal) and server-side (spatio), to mitigate catastrophic forgetting and data heterogeneity; 3) We leverage prototypes to approximate task-wise gradients, improving gradient alignment on the client-side. Extensive experiments demonstrate the superiority of our method over existing baselines, particularly in scenarios with a large number of sequential tasks, highlighting its effectiveness in addressing the complexities of real-world FCL.

## 1 Introduction

In Federated Continual Learning (FCL), clients collaboratively learn models for their private, sequential tasks while preserving data privacy. However, due to the sequential nature of these tasks, each client only has access to a limited amount of data from the current task (Li et al., 2025b). This constraint often leads to the loss of previously acquired knowledge, resulting in catastrophic forgetting. The challenge becomes even more pronounced in heterogeneous FCL (Wuerkaixi et al., 2024), where the clients are engaged in non-identical tasks at any given time, resulting in a non-uniform learning environment. Specifically, the model suffers from both catastrophic forgetting and client drift, which causes negative transfer from the client's current tasks to the previous tasks and other clients' tasks, respectively. Our empirical analysis reveals that existing FCL methods fail to adequately address these issues. Most approaches focus solely on mitigating catastrophic forgetting at the client level, while overlooking the generalization of the global model (see Figure 1). Other methods (Zhang et al., 2023b; Tran et al., 2024) attempt to share knowledge among clients by training a generative model at the server to produce synthetic data for clients. However, broadcasting such synthetic data to all clients introduces significant communication overhead, which can severely limit the scalability and efficiency of the federated system. Acknowledging these challenges, we take a different perspective:

> *Rather than viewing task heterogeneity as a limitation, can we leverage the diverse tasks across clients to improve generalization in FCL, while maintaining communication efficiency?*

Our intuition is straightforward. Temporal tasks (arising from different time steps within a client) and spatio tasks (arising from heterogeneous clients) can both be viewed as distinct tasks. If an invariant

gradient trajectory can be identified across these tasks, it may guide the model toward improved generalization. Such generalization across heterogeneous tasks can, in turn, promote both stability (by maintaining performance on past tasks) and plasticity (by leveraging diverse spatio tasks), thereby achieving a more balanced and effective FCL system.

From these intuitions, we propose a novel method, dubbed Federated Continual Learning via **S**patio-**T**emporal gr**A**dient align**M**ent with **P**rototypical Coreset (STAMP). In our design, we apply gradient alignment across both spatio and temporal dimensions of the FCL system. By aligning gradients along these two dimensions, STAMP identifies aggregated gradients that minimize negative transfer both across sequential tasks and between clients, thereby improving the generalization ability of the global model. In STAMP, the utilization of temporal gradient alignment requires access to gradients from both current and previous tasks on the current model. However, straightforward approach of storing raw gradients in memory (Luo et al., 2023; Saha et al., 2021; Deng et al., 2021) is insufficient for gradient alignment, as it only preserves past gradients tied to specific tasks and lacks robustness for FCL.

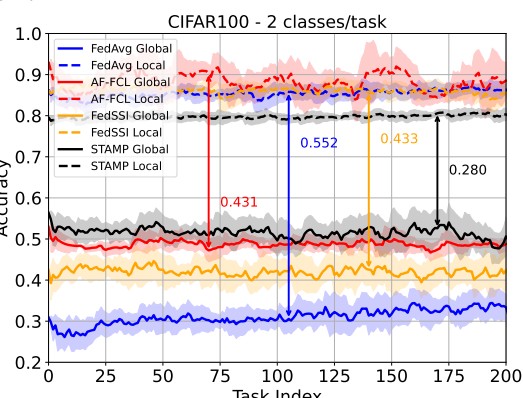

To overcome this limitation, we propose a prototypical coreset selection strategy, in which a compact set of representative data points is stored and subsequently used to construct prototypes. Prototypes provide stable and invariant representations of task-wise gradients (Lv et al., 2022), thereby enabling more reliable gradient alignment. To further enhance the representational power of the prototypes, we employ a prototypical network that ensures accurate prototypes for the prototypical coreset selection even under data perturbations. This approach offers two key advantages. (1) By carefully selecting a compact set of representative samples (coresets), our method maintains prototype quality and diversity over time with significantly reduced dependence on the prototypical networks or generative replay mechanisms used in prior work (Wei et al., 2023; Li et al., 2024a; Chen et al., 2023; Goswami et al., 2023; Qi et al., 2023; Zhen et al., 2020), both of which are vulnerable to catastrophic forgetting. (2) Unlike traditional coreset selection methods that aim to capture the most representative data, our approach focuses on selecting just enough information to ensure

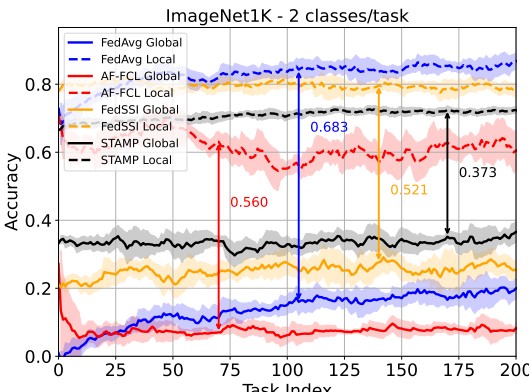

Figure 1: We evaluate leading FCL methods (e.g., AF-FCL (Wuerkaixi et al., 2024), FedSSI (Li et al., 2025c)) under heterogeneous settings and observe a notable gap between local and global test accuracy. These methods exhibit strong personalized performance, as reflected in their high accuracy on local test data. However, their low accuracy on global test data demonstrates limited generalization capability in heterogeneous FCL environments. This limited generalization further indicates insufficient model plasticity when adapting to both previously and new unseen tasks. Our proposed method STAMP shows state-of-art robustness by mitigating inter-client divergence throughout the learning process, leading to a reduced local-global generalization gap.

stable prototype estimation. This enables our system to store significantly fewer samples per class compared to prior methods, while still maintaining sufficient information to approximate gradient trajectories for reliable gradient alignment.

## 2 BACKGROUNDS & PRELIMINARIES

### 2.1 FEDERATED CONTINUAL LEARNING

FCL refers to a practical learning scenario that melds the principles of FL and CL. Suppose that there are $U$ clients. On each client $u$, the model is trained on a sequence of $T$ tasks. At a given step $t \times R + r$, where $R$ represents the number of communication rounds per task and $r$ is the current round of task $t$, client $u$ holds model parameters $\theta_u^{t,r}$ and only has access to the data from task $t$. On client $u$, data $\mathcal{D}_u^t$ of task $t$ consists of $N_u^t$ pairs of samples and their labels, i.e., $\mathcal{D}_u^t = \{(x_i^t, y_i^t)_{i=1}^{N_u^t}\}$.

In existing literature, the primary focus is on a specific task reshuffling setting, wherein the task set is identical for all clients, yet the arrival sequence of tasks differs (Yoon et al., 2021). In practical scenarios, it may be observed that the task set of clients is not necessarily correlated. There is no guaranteed relation among the tasks $\{\mathcal{D}_u^1, \mathcal{D}_u^2, \ldots, \mathcal{D}_u^T\}$ of client $u$ at different steps. Similarly, there is no guaranteed relation among the tasks $\{\mathcal{D}_1^t, \mathcal{D}_2^t, \ldots, \mathcal{D}_U^t\}$ across different clients. Thus, we consider a more practical setting, the Limitless Task Pool (LTP).

**Limitless Task Pool.** In the setting of LTP, tasks are selected randomly from a substantial repository of tasks, creating a situation where two clients may not share any common tasks (i.e., $\{\mathcal{D}_u^i\}_{i=1}^{t_u} \cap \mathcal{D}_v^i\}_{i=1}^{t_v} = \varnothing, \forall u, v \in \{1, 2, \ldots, U\}$). More importantly, clients possess diverse joint distributions of data and labels $p(x, y)$ due to statistical heterogeneity. Therefore, features learned from other clients could invariably introduce bias when applied to the current task of a client.

At every task $t$, our goal is to facilitate the collaborative construction of the global model with parameter $\theta^t$. Under the privacy constraint inherent in FL and CL, we aim to harmoniously learn current tasks while preserving performance on previous tasks for all clients, thereby seeking to optimize performance across all tasks seen so far by all clients as follows:

$$\min_{\theta^t} [\mathcal{S}_1^t, \mathcal{S}_2^t, \ldots, \mathcal{S}_U^t], \quad \text{where} \quad S_u^t = [\mathcal{L}(\theta^t; \mathcal{D}_u^1), \mathcal{L}(\theta^t; \mathcal{D}_u^2), \ldots, \mathcal{L}(\theta^t; \mathcal{D}_u^t)]. \tag{1}$$

However, due to the resource limitation of distributed devices, the replay memory on clients are limited. Each client $u$, while performing the task $t$, does not have access to the samples of the previously learned task $\mathcal{D}_u^{[1:t-1]}$. Thus, the client model $\theta_u^t$ cannot be directly optimized to minimize the corresponding empirical risk $\sum_{i=1}^{t} \mathcal{L}(\theta_u^t; \mathcal{D}_u^i)$. Moreover, data heterogeneity on each client at specific task $t$ introduces domain or label shifts, leading to discrepancies in data distributions across tasks and clients. This heterogeneity causes gradient conflict during training (Nguyen et al., 2025).

### 2.2 GRADIENT ALIGNMENT

When learning with various non-identical tasks, gradient conflict is one of the most critical issues.

**Definition 1 (Gradient conflict)** *The gradient $g_i$ and $g_j$ $(i \neq j)$ between two tasks $i, j$ are considered to be in conflict if their cosine similarity is negative, i.e., $\cos(g_i, g_j) = \frac{g_i \cdot g_j}{|g_i| \cdot |g_j|} < 0$. In this scenario, progress along the gradient $g_i$ results in negative transfer with respect to $g_j$, and vice versa.*

To mitigate the gradient conflict among tasks as in Definition 1, we leverage the Gradient Alignment (GA) approach proposed in (Nguyen et al., 2025) to achieve this objective

$$\text{GA}(\mathbf{g}^{(r)}) = \bar{g}^{(r)} + \frac{\kappa \|\bar{g}^{(r)}\|}{\|\Gamma^* \mathbf{g}^{(r)}\|} \Gamma^* \mathbf{g}^{(r)}, \text{s.t. } \Gamma^* = \arg\min_\Gamma \Gamma \mathbf{g}^{(r)} \cdot \bar{g}^{(r)} + \kappa \|\bar{g}^{(r)}\| \|g_\Gamma^{(r)}\|, \tag{2}$$

where $\mathbf{g}^{(r)} = [g_t^{(r)} | t \in \mathcal{T}]$ are the set of task-wise gradients, $\bar{g}^{(r)} = \sum_{t \in \mathcal{T}} \frac{g_t^{(r)}}{|\mathcal{T}|}$ is the averaged gradient over set of tasks $\mathcal{T}$. The learned gradient $g_G = \text{GA}(\mathbf{g}^{(r)})$ utilizes the gradients of multiple tasks $\mathbf{g}^{(r)} = [g_t^{(r)} | t \in \mathcal{T}]$ to preserve the invariant properties of individual task-specific gradients. Specifically, since $g_G$ satisfies the condition $g_G \cdot g_i \geq 0, \forall i \in T$, it ensures that the resulting gradient does not induce negative transfer across tasks. Consequently, the aggregated gradient facilitates generalization across all tasks within the CL framework. The formal proof of the gradient alignment update rule is provided in Appendix B.

## 3 PROPOSED METHOD

We propose a novel framework, STAMP, for heterogeneous FCL. At its core, STAMP involves a gradient alignment on both temporal and spatio tasks to both improving the plasticity while guarantee the stability. Additionally, replay memory with prototypical exemplars is introduced to reduce the memory cost while improving the stability of task-wise gradient approximation.

### 3.1 SPATIO-TEMPORAL GRADIENT ALIGNMENT

**Motivation.** In FCL under heterogeneous settings (Wuerkaixi et al., 2024), the challenges become particularly severe due to the diversity of tasks and data distributions across clients. A major difficulty arises from the inherent communication constraints, which make direct sharing of data or model parameters between clients impractical. Consequently, handling heterogeneous tasks in FCL has remained a largely intractable problem. In this work, we are motivated by drawing an analogy between spatio and temporal tasks in FCL. Specifically, we conceptualize the heterogeneous tasks across clients as a joint composition of spatio and temporal tasks. More importantly, rather than focusing solely on mitigating catastrophic forgetting and client heterogeneity, we investigate the generalization capability of heterogeneous FCL systems through the lens of the generalization gap.

**Theorem 1** *Let $\mathcal{H}$ be a hypothesis space of VC-dimension $M$, $d_{\mathcal{H}\triangle\mathcal{H}}(\mathcal{D}_u^i, \mathcal{D}_v^i)$ is the spatio divergence between clients $u, v$ at task $i$, $d_{\mathcal{H}\triangle\mathcal{H}}(\mathcal{D}_u^i, \mathcal{D}_u^j)$ is the temporal divergence of client $u$ at two different tasks $i, j$. Let $\mathcal{D}_{\mathcal{P}} = \{\mathcal{D}_u^i, \forall i \in [1:t], u \in \mathcal{U}\}$ as the dataset of seen tasks, and $\mathcal{D}_{\mathcal{Q}} = \mathcal{D}\backslash\mathcal{D}_{\mathcal{P}}$ as the dataset of unseen task. For any $\delta \in (0, 1)$, the generalization gap on an unseen task $\mathcal{D}_{\mathcal{Q}}$ is bounded by the following with a probability of at least*

$$1-\delta: \mathcal{E}(\theta; \mathcal{D}_{\mathcal{Q}}) \leq \sum_{i\in\mathcal{T}}\sum_{u\in\mathcal{U}} \gamma_u \Bigg[\mathcal{E}(\theta;\mathcal{D}_u^i) + \sum_{j\in\mathcal{T}} d_{\mathcal{H}\triangle\mathcal{H}}(\mathcal{D}_u^i, \mathcal{D}_u^j) + \sum_{v\in\mathcal{U}} d_{\mathcal{H}\triangle\mathcal{H}}(\mathcal{D}_u^i, \mathcal{D}_v^i) +$$

$$\sqrt{\frac{\log M + \log\frac{1}{\delta}}{2N_u}}\Bigg] + \zeta^*, \text{ where } \zeta^* \text{ is the optimal combined risk on } \mathcal{D}_{\mathcal{P}}, \mathcal{D}_{\mathcal{Q}}, \text{ respectively.}$$

From the Theorem 1, we can see that, to improve the generalization of the FCL system on the unseen task, it is crucial to minimize the temporal divergence $d_{\mathcal{H}\triangle\mathcal{H}}(\mathcal{D}_u^i, \mathcal{D}_u^j)$ and spatio divergence $d_{\mathcal{H}\triangle\mathcal{H}}(\mathcal{D}_u^i, \mathcal{D}_v^i)$. Current works focus on minimizing the $d_{\mathcal{H}\triangle\mathcal{H}}(\mathcal{D}_u^i, \mathcal{D}_u^j)$ among the seen classes $[1:t], t \in \mathcal{T}$ and not efficiently minimize the gap among the clients $d_{\mathcal{H}\triangle\mathcal{H}}(\mathcal{D}_u^i, \mathcal{D}_v^i)$. This is because the minimization of $d_{\mathcal{H}\triangle\mathcal{H}}(\mathcal{D}_u^i, \mathcal{D}_v^i), \forall u, v \in \mathcal{U}$ requiring the knowledge transfer among clients. Recent works (Zhang et al., 2023b; Tran et al., 2024) attempt to solve this challenge by generating synthetic data on the server at each communication round and broadcasting it back to the clients. While this approach enables partial alignment across clients, it incurs substantial communication overhead, which significantly limits the scalability of FCL in large-scale deployments.

To jointly minimize both temporal and spatio divergences, we focus on leveraging gradients across temporal and spatio tasks. This strategy eliminates the need for explicit knowledge transfer between clients and the server, making it highly communication-efficient. Our primary objective is to identify an invariant gradient trajectory that remains stable across both temporal and spatio tasks. By aligning gradients

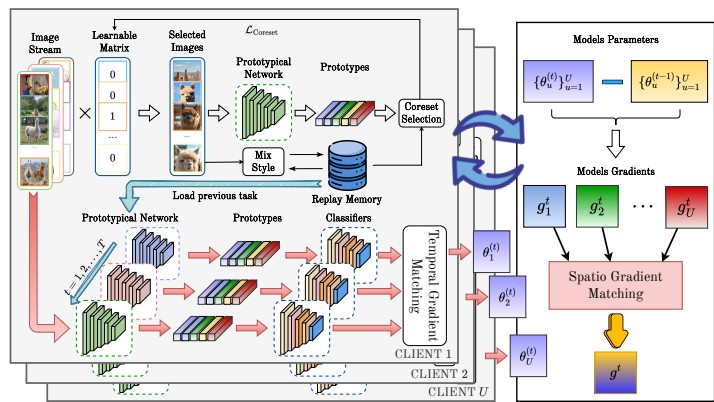

Figure 2: Illustration of STAMP architecture.

in this manner, the learned model can generalize effectively to previously unseen tasks (Shi et al., 2022). From a generalization perspective, improving the model's ability to generalize to unseen

tasks enhances its plasticity, enabling it to more effectively adapt to new tasks without significant performance degradation. Moreover, such generalization inherently mitigates catastrophic forgetting, as the model becomes less prone to overfitting to newly encountered tasks while retaining knowledge from previously learned ones.

**Temporal Gradient Alignment.**  The temporal gradient alignment technique is implemented on the client side in the local training. In particular, we take the gradients of previous tasks as input data for the gradient alignment optimization problem as follows: $\theta_u^{t,r+1} = \theta_u^{t,r} - \text{GA}(\mathbf{g}_u^{[0:t]})$, where $\mathbf{g}_u^{[0:t]} = [g_u^i \mid i = \{1, 2, \ldots, t\}]$ denotes the set of task-specific gradients, including the gradients of previous tasks $\mathbf{g}_u^{[0:t-1]}$ and current task $g_u^t$. Traditionally, the gradients from previous tasks are computed using stored data samples from past tasks to approximate the true gradients (Lopez-Paz & Ranzato, 2017; Luo et al., 2023; Wu et al., 2024). However, this approach requires a substantial memory buffer to store a sufficient amount of data for accurate gradient estimation. In scenarios where storage capacity is limited, the precision of the gradient approximation may be significantly compromised. An alternative solution to compute gradients is via prototype as follows:

$$g_u^{(t)} = \frac{1}{C} \sum_{c=1}^{C} \nabla_{\theta_u^{t,r,E}} \mathcal{L}\Big( f(p_{u,c}^t; \theta_u^{t,r,E}); c \Big). \tag{3}$$

To efficiently compute the prototypes for the gradient estimation, we employ the prototypical network (Snell et al., 2017). However, the prototypical network and its continual counterpart (Wei et al., 2023) may still suffer from catastrophic forgetting when deployed in the CL system. To mitigate this challenge, our intuition is to design prototypes that are learned without relying on prototype networks. To do so, we leverage a prototypical coreset which stores meaningful features for the prototype measurements in CL. The details of the prototypical coreset and its selection method are demonstrated in Section 3.2.

**Spatio Gradient Alignment.**  Building upon the work of (Nguyen et al., 2025), the spatial gradient is computed on the server to identify a consistent gradient direction that remains invariant across heterogeneous tasks in FCL. This facilitates the global model in establishing a stable gradient direction, thereby mitigating the negative transfer that can occur due to task diversity. The update is given as follows:

$$\theta^{t,r+1} = \theta^{t,r} - \text{GA}(\mathbf{g}^t), \quad \mathbf{g}^t = [g_u^t \mid u = \{1, 2, \ldots, U\}], \tag{4}$$

where $\mathbf{g}^t$ represents the collection of local gradients obtained from the participating clients. Each local gradient is computed as $g_u^t = \theta_u^{t,r+1} - \theta_u^{t,r}$, using the model updates, and thus incurs no additional communication overhead. By aligning the gradient directions across clients, this method effectively addresses task heterogeneity, reducing the detrimental impact of client drift in heterogeneous FCL.

## 3.2 Prototypical Coreset assisted Replay Memory

**Prototypical Coreset Selection.**  For each class $l$, our objective is to identify salient set of samples such that their combined representations, as processed by the encoder $\phi$ form a prototype on class $l$. At each task $t$, when we observe data $\mathcal{N}_l^t$ of label $l$, we select a subset $\widetilde{X}^l$ as follows:

$$\widetilde{X}^l = \arg\min_A \left\| \left[ \frac{1}{|\mathcal{M}^l|} \sum_{i \in \mathcal{M}^l} g(x_i; \phi) + \frac{1}{|\mathcal{N}_l^t|} \sum_{i \in \mathcal{N}_l^t} a_i \cdot g(x_i; \phi) \right] - p^l \right\|^2, \tag{5}$$

$$\text{s.t.} \quad p^l = \frac{1}{\sum_{t=1}^T |\mathcal{N}_l^t|} \left[ g(\widetilde{x}^l; \phi) \cdot \sum_{j=1}^{t-1} |\mathcal{N}_l^j| + \sum_{i \in \mathcal{N}_l^t} g(x_i; \phi) \right] \cdot \mathbb{1}\{y_j = l\},$$

$$X^l = \{x_i \mid a_i \in A\}, \ |X^l| = |\mathcal{M}^l|.$$

Here, $\mathcal{M}^l$ is the replay memory for class $l$, with pre-defined memory budget $|\mathcal{M}^l|$. If the number of selected samples exceeds $|\mathcal{M}^l|$, we apply MixStyle to blend the style of the newly selected data with

that of the previously identified samples, as formulated below:

$$\text{MixStyle}(\widetilde{x}^l; x) = \gamma_{\text{mix}} \frac{\widetilde{x}^l - \mu(\widetilde{x}^l)}{\sigma(\widetilde{x}^l)} + \beta_{\text{mix}}, \tag{6}$$

$$\text{s.t.} \quad \gamma_{\text{mix}} = \lambda\sigma(\widetilde{x}^l) + (1-\lambda)\sigma(x), \quad \beta_{\text{mix}} = \lambda\mu(\widetilde{x}^l) + (1-\lambda)\mu(x),$$

where $x$ are the newly satisfying prototypical exemplars found from (6). To make the encoder $\phi$ learn the prototype better, we inherit the prototypical network (Snell et al., 2017) learning process to learn the encoder $\phi$.

**Prototypical Network with Coreset.** On each client $u$, the prototype $p_{u,l}^t$ on label $l$ are computed via a prototypical network (Snell et al., 2017) via $p_{u,l}^t = \frac{1}{|\mathcal{D}_{u,l}^t|} \sum_{x_i \in \mathcal{D}_{u,l}^t} g(x_i; \phi)$. The prototypical network is learned via a loss function as follows:

$$\phi^* = \arg\min_{\phi} \sum_{l=1}^{L} d\Big(g(x; \phi), p_l\Big) - \log \sum_{l'} \exp\Big(d\Big(g(x; \phi), p_l\Big)\Big). \tag{7}$$

The objective of (7) is to ensure that the learned prototype $g(x; \phi)$, derived from the input data $x$, closely aligns with the computed prototype of the same class $l$ across the entire batch, while simultaneously maintaining a significant distance from approximated prototypes of other classes $l'$.

## 4 THEORETICAL ANALYSIS

To conduct the theoretical analysis of STAMP, we examine the generalization gap between the model trained at a specific round $R$ and the model trained on the unseen task dataset $\mathcal{D}_{\mathcal{Q}}$.

**Theorem 2** *Let $\theta^R$ denote the global model after $R$ rounds and at current task $t \in \mathcal{T}$, $\theta_u^*$ and $\theta_{\mathcal{Q}}^*$ mean the optimal of the model on each client and the unseen tasks, respectively. The local objectives follow the $\mu$ strongly convex from Assumption 2. For any $\delta \in (0,1)$, the generalization gap for the unseen tasks $\mathcal{D}_{\mathcal{Q}}$ can be bounded by the following equation with a probability of at least $1 - \delta$:*

$$\mathcal{E}_{\mathcal{D}_{\mathcal{Q}}}(\theta^R) - \mathcal{E}_{\mathcal{D}_{\mathcal{Q}}}(\theta_{\mathcal{Q}}^*) \leq \sum_{i \in [1:t]} \sum_{u \in \mathcal{U}} \gamma_u \zeta_t \Bigg[ \mathcal{E}_{\hat{\mathcal{D}}_u}(\theta) + \sum_{j \in [1:t]} \frac{d_{\mathcal{G} \circ \theta}(\hat{\mathcal{D}}_u^i, \hat{\mathcal{D}}_u^j)}{\mu} + \sum_{v \in \mathcal{U}} \frac{d_{\mathcal{G} \circ \theta}(\hat{\mathcal{D}}_u^i, \hat{\mathcal{D}}_v^i)}{\mu}$$

$$+ d_{\mathcal{H} \triangle \mathcal{H}}(\mathcal{D}_{\mathcal{P}}, \mathcal{D}_{\mathcal{Q}}) + \frac{\sqrt{\log \frac{M}{\delta}} + \sqrt{\log \frac{UM}{\delta}}}{\sqrt{2N_u}} \Bigg] + \zeta^*, t \in \mathcal{T},$$

*where $\hat{\mathcal{D}}_u^i, \hat{\mathcal{D}}_u^j, \hat{\mathcal{D}}_v^i$ are the sampled counterparts from the domain $\mathcal{D}_u^i, \mathcal{D}_u^j, \mathcal{D}_u^i$, respectively. $d_{\mathcal{G} \circ \theta}(\hat{\mathcal{D}}_u^i, \hat{\mathcal{D}}_u^j)$ denotes the gradient divergence when training on temporal tasks $\hat{\mathcal{D}}_u^i$ and $\hat{\mathcal{D}}_u^j$. $d_{\mathcal{G} \circ \theta}(\hat{\mathcal{D}}_u^i, \hat{\mathcal{D}}_v^i)$ denotes the gradient divergence when training on spatio tasks $\hat{\mathcal{D}}_u^i$ and $\hat{\mathcal{D}}_v^i$.*

In contrast to existing studies on convergence in FCL (Keshri et al., 2025), our work focuses on establishing theoretical guarantees for the generalization gap. This generalization perspective enables a principled assessment of how reliably an FCL model can extend to both previously encountered and new unseen tasks, thereby characterizing its stability and plasticity.

The generalization gap at round $R$ on the target domain is defined as $\mathcal{E}_{\mathcal{D}_{\mathcal{Q}}}(\theta^R) - \mathcal{E}_{\mathcal{D}_{\mathcal{Q}}}(\theta_{\mathcal{D}_{\mathcal{Q}}}^*)$. In Theorem 2, the first term $\mathcal{E}_{\hat{\mathcal{D}}_u}(\theta)$ is the loss on the local datasets. The fourth term $d_{\mathcal{H} \triangle \mathcal{H}}(\mathcal{D}_{\mathcal{P}}, \mathcal{D}_{\mathcal{Q}})$ is the task divergence between the seen and unseen tasks. The fifth term $\frac{\sqrt{\log \frac{M}{\delta}} + \sqrt{\log \frac{UM}{\delta}}}{\sqrt{2N_u}}$ is the gap due to the infinite sampling. The last term $\zeta^*$ is the gap due to the optimal risk. While the first term is the main minimization on every FCL methods, the three last terms are irreducible. To further reduce the generalization gap, our objective is to minimize this gradient divergence at each round. Specifically, STAMP focuses on reducing the temporal gradient divergence $\sum_{j \in [1:t]} \frac{d_{\mathcal{G} \circ \theta}(\hat{\mathcal{D}}_u^i, \hat{\mathcal{D}}_u^j)}{\mu}$, and spatio gradient divergence $\sum_{v \in \mathcal{U}} \frac{d_{\mathcal{G} \circ \theta}(\hat{\mathcal{D}}_u^i, \hat{\mathcal{D}}_v^i)}{\mu}$, using spatio and temporal gradient alignment every server aggregation round. Following Appendix B, we have the temporal and spatio gradient

divergence are minimized over the gradient alignment. As a consequence, we can directly reduce those gap. By effectively leveraging STAMP, we can reduce the generalization gap between seen and unseen tasks, thereby enhancing the overall generalization capability of the heterogeneous FCL system.

## 5 EXPERIMENTAL RESULTS

In this section, we conduct extensive experiments to demonstrate the effectiveness of STAMP. The implementation details and additional experiments are provided in Appendices E . To ensure a fair assessment of FCL baselines under heterogeneous settings and catastrophic forgetting, we do not use pretrained models, as their training data (e.g., ImageNet1K) overlaps with our dataset, potentially biasing the evaluation. The detailed configurations of the continual data settings, model settings, and baseline setups are provided in Appendix E.

### 5.1 BENCHMARKING

Table 1: We report the average per-task performance of FCL under a setting where each task is assigned 2 classes. Evaluations are conducted using 10 clients (fraction = 1.0) across 5 independent trials. OOM refers to the out of memory in GPU. ↑ and ↓ indicate that higher and lower values are better, respectively. C→S and S→C denote communication from the client to the server and from the server to the client, respectively.

| Methods | Accuracy ↑ | AF ↓ | Avg. Comp. ↓ (Sec/Round) | Comm. Cost ↓ C→S | S → C | GPU (Peak) ↓ | Disk ↓ |
|---|---|---|---|---|---|---|---|
| **S-CIFAR100** ($U = 10$, $C = 2$) | | | | | | | |
| FedAvg | 31.7 (± 1.7) | 25.2 (± 1.3) | 3.3 sec | 44.6 MB | 44.6 MB | 1.92 GB | N/A |
| FedDBE | 37.0 (± 1.6) | 26.1 (± 0.7) | 3.6 sec | 44.6 MB | 44.6 MB | 1.91 GB | N/A |
| FedAS | 58.2 (± 0.1) | 56.1 (± 0.1) | 13.7 sec | 44.6 MB | 44.6 MB | 1.92 GB | N/A |
| FedOMG | 39.1 (± 1.3) | 24.5 (± 0.4) | 4.1 sec | 44.6 MB | 44.6 MB | 1.92 GB | N/A |
| GLFC | 44.8 (± 2.1) | 29.5 (± 0.4) | 18.3 sec | 88.2 MB | 46.5 MB | 4.33 GB | 22.1 MB |
| FedCIL | 46.5 (± 2.2) | 28.8 (± 1.2) | 22.3 sec | 95.3 MB | 44.6 MB | 4.81 GB | 18.5 MB |
| LANDER | 50.8 (± 1.3) | 26.9 (± 0.4) | 15.8 sec | 88.2 MB | 104.3 MB | 5.26 GB | 131.5 MB |
| TARGET | 45.1 (± 2.4) | 28.6 (± 1.6) | 25.6 sec | 112.4 MB | 44.6 MB | 3.65 GB | 18.5 MB |
| FedL2P | 48.2 (± 1.8) | 28.1 (± 0.6) | 8.6 sec | 56.3 MB | 56.3 MB | 2.56 GB | N/A |
| Re-Fed+ | 52.3 (± 1.1) | 31.9 (± 0.5) | 3.9 sec | 44.6 MB | 44.6 MB | 2.17 GB | 18.5 MB |
| FedWeIT | 52.6 (± 1.3) | 25.7 (± 0.9) | 5.4 sec | 44.6 MB | 44.6 MB | 5.83 GB | 61.7 GB |
| FedSSI | 51.6 (± 1.3) | 35.4 (± 1.1) | 7.7 sec | 44.6 MB | 44.6 MB | 2.53 GB | N/A |
| AF-FCL | 51.4 (± 0.7) | 48.7 (± 1.2) | 4.9 sec | 156.3 MB | 121.3 MB | 8.93 GB | N/A |
| **STAMP** | 52.8 (± 0.9) | 24.3 (± 0.8) | 9.1 sec | 44.6 MB | 44.6 MB | 1.92 GB | 16.3 MB |
| **S-ImageNet1K** ($U = 10$, $C = 2$) | | | | | | | |
| FedAvg | 24.3 (± 5.1) | 19.6 (± 0.1) | 133.2 sec | 112.5 MB | 112.5 MB | 16.11 GB | N/A |
| FedDBE | 29.2 (± 7.2) | 19.4 (± 0.2) | 142.7 sec | 112.5 MB | 112.5 MB | 16.11 GB | N/A |
| FedAS | 43.5 (± 4.4) | 40.2 (± 0.4) | 498.5 sec | 112.5 MB | 112.5 MB | 16.11 GB | N/A |
| FedOMG | 30.4 (± 3.8) | 21.1 (± 0.7) | 171.3 sec | 112.5 MB | 112.5 MB | 16.11 GB | N/A |
| GLFC | 31.4 (± 3.1) | 27.4 (± 0.6) | 466.7 sec | 225.3 MB | 121.2 MB | 20.24 GB | 221.4 MB |
| FedCIL | 33.8 (± 3.6) | 25.8 (± 0.7) | 652.3 sec | 245.5 MB | 112.5 MB | 23.47 GB | 184.3 MB |
| LANDER | 34.9 (± 2.7) | 26.1 (± 0.9) | 573.8 sec | 267.4 MB | 453.6 MB | 26.54 GB | 1.31 GB |
| TARGET | 33.2 (± 4.2) | 25.2 (± 0.4) | 913.2 sec | 287.4 MB | 112.5 MB | 21.08 GB | 184.3 MB |
| FedL2P | 34.5 (± 4.8) | 26.4 (± 0.2) | 303.7 sec | 146.6 MB | 146.6 MB | 18.21 GB | N/A |
| Re-Fed+ | 35.3 (± 0.7) | 26.1 (± 1.0) | 146.8 sec | 112.5 MB | 112.5 MB | 16.71 GB | 184.3 MB |
| FedWeIT | 39.7 (± 3.1) | 21.5 (± 0.9) | 194.2 sec | 111.8 MB | 111.8 MB | 62.7 GB | 640 GB |
| FedSSI | 38.4 (± 1.2) | 31.9 (± 0.8) | 298.1 sec | 112.5 MB | 112.5 MB | 17.66 GB | N/A |
| AF-FCL | 38.3 (± 5.3) | 36.6 (± 0.3) | 176.7 sec | 421.3 MB | 336.8 MB | 46.81 GB | N/A |
| **STAMP** | 41.5 (± 2.4) | 24.2 (± 0.8) | 321.2 sec | 112.5 MB | 112.5 MB | 16.11 GB | 152.6 MB |

**Main Results.** Table 1 reports results on the S-CIFAR100 dataset (Boschini et al., 2022) and the S-ImageNet1K dataset (Dohare et al., 2024), which are continual learning versions of CIFAR100 and ImageNet1K. In these settings, each task comprises two distinct classes. In addition to average accuracy and average forgetting (AF), we assess key system-level metrics: computational overhead, communication cost, GPU utilization, and disk usage. Computational overhead is measured as the average time per round, reflecting the cost of client-side training, especially for generative models. Communication cost denotes the average data transferred (in GB) per client-server round. GPU utilization captures peak memory usage, critical in resource-limited settings, while disk usage reflects the total client-side storage required, including replay buffers and task-specific model parameters. The vanilla FL baselines, e.g., FedAvg, FedAS, FedDBE, and FedOMG, may lead the model easily to forget the knowledge from past tasks, as indicated by high average forgetting.

FedWeIT[1] stores task-specific head parameters in GPU memory. However, when both the number of classes (e.g., 1000 classes in S-ImageNet1K) and the number of tasks (e.g., 500 tasks in our S-ImageNet1K setup) become large, the total number of parameters grows significantly[2]. As a result, storing all task-specific parameters in GPU memory becomes infeasible, and they must instead be saved to disk. However, this approach leads to a substantial increase in average training time. LANDER stores all generated pseudo task-specific data on disk, incurring client-side storage overhead comparable to conventional CL methods using replay memory. Additionally, broadcasting synthetic data from the server to clients introduces substantial communication overhead.

The key observations from Tables 1 indicate that the more challenging setting, with only two classes per task, exhibit greater susceptibility to catastrophic forgetting. This is because each task provides less comprehensive information about the overall dataset, thereby leading to a higher average forgetting (AF) score. STAMP achieves the state-of-art overall trade-off, delivering higher accuracy and lower forgetting than almost all methods. At the same time, STAMP communication cost remains comparable to that of standard FL and requires relatively modest RAM and disk resources. For example, it is worth noting that FedWeIT achieves slightly lower forgetting on S-CIFAR100 at the expense of **nearly 3900× higher disk usage**. This making STAMP suitable for deployment on resource-constrained devices. To ensure comparability with other popular works in FCL, we also evaluate the benchmark on an easier class distribution, where each task contains 20 distinct classes. The corresponding results are reported in appendix F.1.

**Performance under tasks with non-IID settings.** Figure 3 illustrates the test accuracy across varying levels of data heterogeneity for CIFAR10, CIFAR100, Digit10, and Office31 datasets. As shown in the figure, all methods improve test accuracy as data heterogeneity decreases (i.e., larger $\alpha$). Notably, STAMP consistently achieves superior and stable performance across different levels of heterogeneity, indicating its robustness under non-IID conditions.

## 5.2 EXPERIMENTAL ANALYSES AND ABLATION TESTS

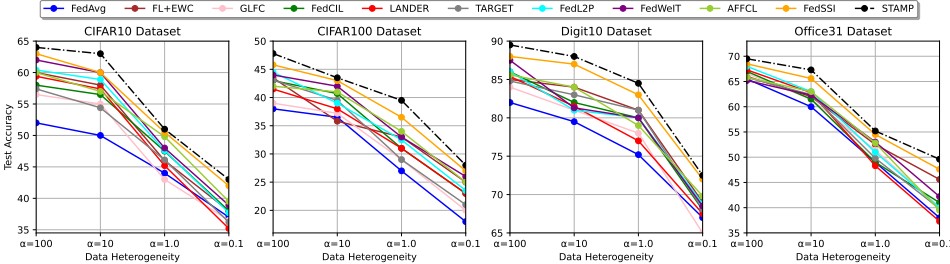

Figure 3: Performance w.r.t data heterogeneity $\alpha$ for four datasets.

**Efficiency of Prototypical Coreset.** To evaluate the effectiveness of our proposed coreset selection method, we compare STAMP with a vanilla FL framework incorporating alternative data condensation techniques on the client side, including SRe²L (Yin et al., 2023), BCSR (Hao et al., 2023), and OCS

---

[1]The official code of FedWeIT can be found at: https://github.com/wyjeong/FedWeIT.

[2]We observe from the official code that FedWeIT needs more than 512 GB of RAM memory to be able to run a simple LeNet on ImageNet. As such, we have to save the task-adaptive parameters in memory. In our reformatted implementation, we mitigate this memory constraint by utilizing disk storage for model loading.

(Yoon et al., 2022), CSReL (Tong et al., 2025). The experimental results in Figure 4 show that our method consistently outperforms these coreset selection-based FL algorithms. Notably, our approach can reduce the coreset size to as few as 20 images per class without significantly compromising performance compared to training on the full-scale dataset for previous tasks.

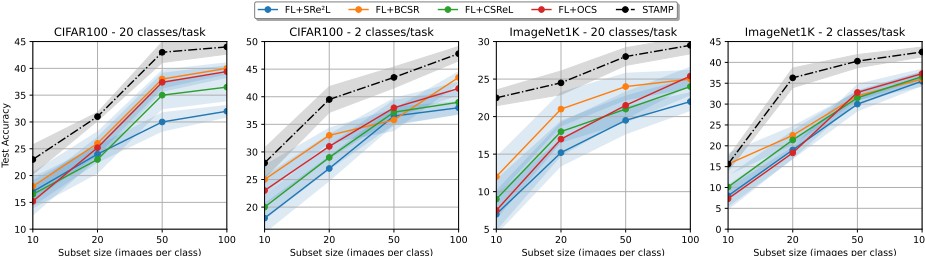

Figure 4: Performance comparisons in coreset selection demonstrate that our approach outperforms the integration of alternative baseline methods within vanilla FL.

**Efficiency of Temporal Gradient Alignment.** To evaluate the effectiveness of temporal gradient alignment on the client side, we analyze the gradient angles produced by STAMP on S-CIFAR100 and S-ImageNet1K datasets and compare them with two sets of baseline methods: FedAvg and FedL2P for standard FL, and FedWeIT and AF-FCL, for FCL. The results are presented in Figure 5. As shown, STAMP demonstrates superior gradient alignment with previously learned tasks. This improvement suggests that STAMP is less prone to catastrophic forgetting compared to existing approaches. Additional results linking gradient angles to catastrophic forgetting are provided in Appendix F.3.

**Efficiency of Spatio Gradient Alignment.** Figure 6 presents the gradient divergence across various baseline methods on S-CIFAR100 and S-ImageNet1K, evaluated under two different settings: 20 classes per task and the more challenging 2 classes per task. It is evident that, unlike existing baselines which generally overlook the alignment among client gradients, STAMP achieves significantly better gradient alignment. This improved alignment facilitates model updates that more effectively seek invariant aggregated gradient directions across clients for specific tasks, thereby enhancing the generalization capability of the aggregated model. This observation is consistent with the reduced global-local generalization gap demonstrated in Figure 1.

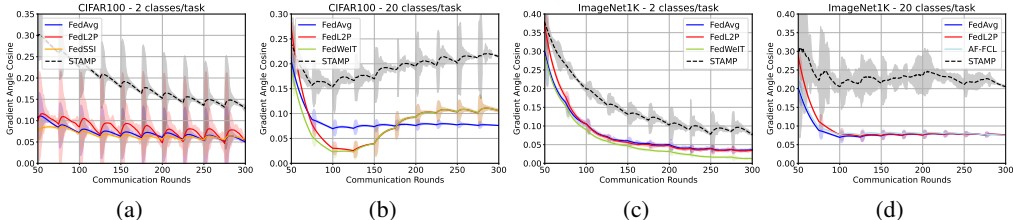

Figure 5: The figures illustrate the average temporal gradient angles across different baseline methods. Specifically, Figure 5a shows the results for S-CIFAR100 under 2 classes per task. Figure 5b shows the gradient cosine similarity on S-CIFAR100 under a 20 classes per task setting. Figure 5c presents the gradient cosine similarity for S-ImageNet1K with 2 classes per task, and Figure 5d depicts the results for S-ImageNet1K under 20 classes per task configuration.

**Ablation Study on STAMP.** Table 2 presents the ablation results for each component. The results demonstrate that both Spatio grAdient alignMent (SAM) and Temporal grAdient alignMent (TAM) consistently enhance the average classification accuracy. Notably, SAM contributes more significantly to accuracy improvement by enhancing generalization across tasks within a single communication round. In contrast, TAM plays a more critical role in reducing average forgetting by mitigating catastrophic forgetting; it achieves this by aligning the learned gradients with those from previous

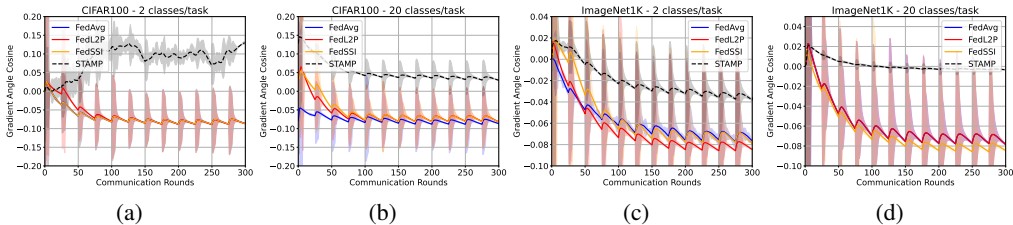

(a)          (b)          (c)          (d)

Figure 6: The figures illustrate the average spatio gradient angles across different baseline methods. Specifically, Figure 6a shows the gradient cosine similarity on S-CIFAR100 under a 2 classes per task setting, Figure 6b shows the results for S-CIFAR100 under 20 classes per task, Figure 6c presents the gradient cosine similarity for S-ImageNet1K with 2 classes per task, and Figure 6d depicts the results for S-ImageNet1K under 20 classes per task configuration.

Table 2: We conduct ablation studies on the S-CIFAR100 and S-ImageNet1K datasets, using 10 clients and 2 classes per task. Specifically, **(1)** refers to spatio-temporal gradient alignment performed on the server side, **(2)** denotes temporal gradient alignment executed on the client side, and **(3)** corresponds to the use of a prototypical coreset implemented with ProtoNet.

| Dataset | Metric | FedAvg | (1) | (2) | (1) + (2) | (1) + (3) | (2) + (3) | STAMP |
|---|---|---|---|---|---|---|---|---|
| S-CIFAR100 | Acc. | 31.7 ($\pm$ 1.7) | 38.1 ($\pm$ 1.3) | 37.8 ($\pm$ 0.6) | 44.7 ($\pm$ 1.5) | 46.1 ($\pm$ 0.7) | 44.9 ($\pm$ 1.4) | 52.8 ($\pm$ 0.9) |
| | AF | 22.1 ($\pm$ 1.3) | 23.8 ($\pm$ 0.4) | 21.7 ($\pm$ 0.9) | 21.5 ($\pm$ 1.0) | 24.7 ($\pm$ 1.4) | 21.8 ($\pm$ 0.6) | 24.3 ($\pm$ 0.8) |
| S-ImageNet1K | Acc. | 24.3 ($\pm$ 5.1) | 30.5 ($\pm$ 2.8) | 28.3 ($\pm$ 2.6) | 34.1 ($\pm$ 0.7) | 37.4 ($\pm$ 1.1) | 36.5 ($\pm$ 1.3) | 41.5 ($\pm$ 2.8) |
| | AF | 19.6 ($\pm$ 0.1) | 26.1 ($\pm$ 0.7) | 23.8 ($\pm$ 0.6) | 24.3 ($\pm$ 0.9) | 26.1 ($\pm$ 1.8) | 23.3 ($\pm$ 0.8) | 24.2 ($\pm$ 0.8) |

tasks on the same client. Additionally, the use of the prototypical coreset selection method further boosts the performance of STAMP by improving data representation through ProtoNet.

## 6 CONCLUSION

In this paper, we have tackled the challenges of FCL in realistic settings characterized by client data heterogeneity and task conflicts. Recognizing the limitations of existing generative replay-based methods, we have introduced a novel model-agnostic approach, Spatio-Temporal Gradient Alignment with Prototypical Coreset. Our method effectively mitigates catastrophic forgetting and data bias by leveraging prototype samples for robust gradient approximation and applying gradient alignment both temporally and spatially. Through extensive experiments, we have demonstrated that our approach consistently outperforms existing baselines, highlighting its potential as a powerful solution for resilient FCL in diverse, dynamic environments.

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

# A PROOF ON THEOREMS

## A.1 TECHNICAL ASSUMPTIONS

**Assumption 1 ($L$-smoothness)** *Each local objective function is Lipschitz smooth, that is,*

$$\|\nabla\mathcal{E}(x;\mathcal{D}_u) - \nabla\mathcal{E}(y;\mathcal{D}_u)\| \leq L\|\mathcal{E}(x;\mathcal{D}_u) - \mathcal{E}(y;\mathcal{D}_u)\|, \forall u \in \mathcal{U}. \tag{8}$$

**Assumption 2 ($\mu$-strongly convex)** *Each local objective function is Lipschitz smooth, that is,*

$$\|\nabla\mathcal{E}(x;\mathcal{D}_u) - \nabla\mathcal{E}(y;\mathcal{D}_u)\| \geq \mu\|\mathcal{E}(x;\mathcal{D}_u) - \mathcal{E}(y;\mathcal{D}_u)\|, \forall u \in \mathcal{U}. \tag{9}$$

**Assumption 3 (Domain triangle inequality (Zhao et al., 2019))** *For any hypothesis space $\mathcal{H}$, it can be readily verified that $d_{\mathcal{H}}(\cdot,\cdot)$ satisfies the triangular inequality:*

$$d_{\mathcal{H}\triangle\mathcal{H}}(\mathcal{D},\mathcal{D}^{''}) \leq d_{\mathcal{H}\triangle\mathcal{H}}(\mathcal{D},\mathcal{D}^{'}) + d_{\mathcal{H}\triangle\mathcal{H}}(\mathcal{D}^{'},\mathcal{D}^{''}). \tag{10}$$

## A.2 TECHNICAL LEMMAS

**Lemma 1 (Task Divergence Decomposition)** *For any hypothesis space $\mathcal{H}$, it can be readily verified that the distance function $d_{\mathcal{H}}(\cdot,\cdot)$ satisfies the triangle inequality. Specifically, for any three distributions $D_u^i, D_v^j, D_u^j$ defined over the same space, we have:*

$$d_{\mathcal{H}}(D_u^i, D_v^j) \leq d_{\mathcal{H}}(D_u^i, D_u^j) + d_{\mathcal{H}}(D_u^j, D_v^j).$$

*Proof.* Applying Assumption 3, we solved the lemma.

**Lemma 2** *If we have $\mathcal{E}_{\hat{\mathcal{D}}}(\theta) = \sum_{u \in \mathcal{U}} \gamma_u \mathcal{E}_{\hat{\mathcal{D}}_u}$, then for any unseen domain $\mathcal{D}_{\mathcal{Q}}$, we have:*

$$d_{\mathcal{H}\triangle\mathcal{H}}(\mathcal{D}_{\mathcal{P}}, \mathcal{D}_{\mathcal{Q}}) = \sum_{u \in \mathcal{U}} \gamma_u d_{\mathcal{H}\triangle\mathcal{H}}(\mathcal{D}_u, \mathcal{D}_{\mathcal{Q}}). \tag{11}$$

*Proof.* From the definition of $d_{\mathcal{H}\triangle\mathcal{H}}(\cdot,\cdot)$ in (Arjovsky et al., 2020), we can get

$$d_{\mathcal{H}\triangle\mathcal{H}}(\mathcal{D}_{\mathcal{P}}, \mathcal{D}_{\mathcal{Q}}) = 2\sup_{A \in \mathcal{A}_{\mathcal{H}\triangle\mathcal{H}}} |\text{Pr}_{\hat{\mathcal{D}}}(A) - \text{Pr}_{\mathcal{D}_{\mathcal{Q}}}(A)| = 2\sup_{A \in \mathcal{A}_{\mathcal{H}\triangle\mathcal{H}}} \Big|\sum_{u \in \mathcal{U}} \gamma_u \text{Pr}_{\hat{\mathcal{D}}}(A) - \text{Pr}_{\mathcal{D}_{\mathcal{Q}}}(A)\Big|$$

$$\leq 2\sup_{A \in \mathcal{A}_{\mathcal{H}\triangle\mathcal{H}}} \Big|\sum_{u \in \mathcal{U}} \gamma_u \Big[\text{Pr}_{\hat{\mathcal{D}}}(A) - \text{Pr}_{\mathcal{D}_{\mathcal{Q}}}(A)\Big]\Big|$$

$$\leq 2\sup_{A \in \mathcal{A}_{\mathcal{H}\triangle\mathcal{H}}} \sum_{u \in \mathcal{U}} \gamma_u |\text{Pr}_{\hat{\mathcal{D}}}(A) - \text{Pr}_{\mathcal{D}_{\mathcal{Q}}}(A)|$$

$$\leq 2\sum_{u \in \mathcal{U}} \gamma_u \sup_{A \in \mathcal{A}_{\mathcal{H}\triangle\mathcal{H}}} |\text{Pr}_{\hat{\mathcal{D}}}(A) - \text{Pr}_{\mathcal{D}_{\mathcal{Q}}}(A)|$$

$$= \sum_{u \in \mathcal{U}} \gamma_u d_{\mathcal{H}\triangle\mathcal{H}}(\hat{\mathcal{D}}_u, \mathcal{D}_{\mathcal{Q}}). \tag{12}$$

**Lemma 3** *For any $\theta \in \Theta$, the expectation risk gap between domain $A$ and domain $B$ is bounded by the domain divergence $d_{\mathcal{H}\triangle\mathcal{H}}(A, B)$.*

$$|\mathcal{E}_A(\theta) - \mathcal{E}_B(\theta)| \leq \frac{1}{2} d_{\mathcal{H}\triangle\mathcal{H}}(A, B). \tag{13}$$

*Proof.* By the definition of $d_{\mathcal{H}\triangle\mathcal{H}}(\cdot,\cdot)$ in (Arjovsky et al., 2020), we have:

$$d_{\mathcal{H}\triangle\mathcal{H}}(A,B) = 2 \sup_{\theta,\theta'\in\Theta} \left| \Pr_{x\sim A}[f(x;\theta) \neq f(x;\theta')] - \Pr_{x\sim B}[f(x;\theta) \neq f(x;\theta')] \right|, \qquad (14)$$

where $f(x;\theta)$ means the prediction function on data $x$ with model parameter $\theta$. We chose $\theta'$ as parameter of the label function, then $f(x;\theta) \neq f(x;\theta')$ means the loss function $\mathcal{L}(x;\theta)$, so we have:

$$d_{\mathcal{H}\triangle\mathcal{H}}(A,B) = 2 \sup_{\theta\in\Theta} \left| \Pr_{x\sim A}[\mathcal{L}(x;\theta)] - \Pr_{x\sim B}[\mathcal{L}(x;\theta)] \right| \geq 2|\mathcal{E}_A(\theta) - \mathcal{E}_B(\theta)|. \qquad (15)$$

Here, $(a)$ holds due to Assumption 1.

**Lemma 4 (Guarantee of inter-client spatio task divergence)** *If we have $\mathcal{E}_{\hat{\mathcal{D}}}(\theta) = \sum_{u\in\mathcal{U}} \gamma_u \mathcal{E}_{\hat{\mathcal{D}}_u}$, then for any domain $\mathcal{D}_\mathcal{P}$, we have:*

$$\sum_{u\in\mathcal{U}} \gamma_u d_{\mathcal{H}\triangle\mathcal{H}}(\hat{\mathcal{D}}_u, \mathcal{D}_\mathcal{P}) \leq \sum_{u\in\mathcal{U}} \sum_{v\in\mathcal{U}} \gamma_u \gamma_v d_{\mathcal{H}\triangle\mathcal{H}}(\hat{\mathcal{D}}_u, \hat{\mathcal{D}}_v). \qquad (16)$$

*Proof.* From the definition of $d_{\mathcal{H}\triangle\mathcal{H}}(\cdot,\cdot)$ in (Arjovsky et al., 2020), we can get

$$\sum_{u\in\mathcal{U}} \gamma_u d_{\mathcal{H}\triangle\mathcal{H}}(\hat{\mathcal{D}}_u, \mathcal{D}_\mathcal{P}) = 2 \sum_{u\in\mathcal{U}} \gamma_u \sup_{A\in\mathcal{A}_{\mathcal{H}\triangle\mathcal{H}}} |\Pr_{\hat{\mathcal{D}}_u}(A) - \Pr_{\mathcal{D}_\mathcal{P}}(A)|$$

$$= 2 \sum_{u\in\mathcal{U}} \gamma_u \sup_{A\in\mathcal{A}_{\mathcal{H}\triangle\mathcal{H}}} |\Pr_{\hat{\mathcal{D}}_u}(A) - \sum_{v\in\mathcal{U}} \gamma_v \Pr_{\hat{\mathcal{D}}_v}(A)|$$

$$= 2 \sum_{u\in\mathcal{U}} \gamma_u \sup_{A\in\mathcal{A}_{\mathcal{H}\triangle\mathcal{H}}} | \sum_{v\in\mathcal{U}} \gamma_v \Pr_{\hat{\mathcal{D}}_u}(A) - \sum_{v\in\mathcal{U}} \gamma_v \Pr_{\hat{\mathcal{D}}_v}(A)|$$

$$\leq 2 \sum_{u\in\mathcal{U}} \gamma_u \sum_{v\in\mathcal{U}} \gamma_v \sup_{A\in\mathcal{A}_{\mathcal{H}\triangle\mathcal{H}}} |\Pr_{\hat{\mathcal{D}}_u}(A) - \Pr_{\hat{\mathcal{D}}_v}(A)|$$

$$\leq \sum_{u\in\mathcal{U}} \sum_{v\in\mathcal{U}} \gamma_u \gamma_v d_{\mathcal{H}\triangle\mathcal{H}}(\hat{\mathcal{D}}_u, \hat{\mathcal{D}}_v). \qquad (17)$$

**Lemma 5 (Guarantee of intra-client temporal task divergence)** *Consider a client $u$, where $\mathcal{D}_u$ is the on-client joint dataset, consisting of $T$ tasks $\mathcal{D}_u = \{\mathcal{D}_u^t | t \in \{1,\ldots,T\}\}$. If we have $\mathcal{E}_{\hat{\mathcal{D}}}(\theta) = \sum_{u\in\mathcal{U}} \gamma_u \mathcal{E}_{\hat{\mathcal{D}}_u}$, then for any domain $\mathcal{D}_\mathcal{P}$, we have:*

$$\sum_{t\in\mathcal{T}} \zeta_t d_{\mathcal{H}\triangle\mathcal{H}}(\hat{\mathcal{D}}_u^t, \hat{\mathcal{D}}_u) \leq \sum_{i\in\mathcal{T}} \sum_{j\in\mathcal{T}} \zeta_i \zeta_j d_{\mathcal{H}\triangle\mathcal{H}}(\hat{\mathcal{D}}_u^i, \hat{\mathcal{D}}_u^j). \qquad (18)$$

*Proof.* From the definition of $d_{\mathcal{H}\triangle\mathcal{H}}(\cdot,\cdot)$ in (Arjovsky et al., 2020), we can get

$$\sum_{t\in\mathcal{T}} \zeta_t d_{\mathcal{H}\triangle\mathcal{H}}(\hat{\mathcal{D}}_u^t, \hat{\mathcal{D}}_u) = 2 \sum_{t\in\mathcal{T}} \zeta_t \sup_{A\in\mathcal{A}_{\mathcal{H}\triangle\mathcal{H}}} |\Pr_{\hat{\mathcal{D}}_u^t}(A) - \Pr_{\hat{\mathcal{D}}_u}(A)|$$

$$= 2 \sum_{i\in\mathcal{T}} \zeta_i \sup_{A\in\mathcal{A}_{\mathcal{H}\triangle\mathcal{H}}} |\Pr_{\hat{\mathcal{D}}_u^i}(A) - \sum_{j\in\mathcal{T}} \zeta_j \Pr_{\hat{\mathcal{D}}_u^j}(A)|$$

$$= 2 \sum_{i\in\mathcal{T}} \zeta_i \sup_{A\in\mathcal{A}_{\mathcal{H}\triangle\mathcal{H}}} | \sum_{j\in\mathcal{T}} \zeta_j \Pr_{\hat{\mathcal{D}}_u^i}(A) - \sum_{j\in\mathcal{T}} \zeta_j \Pr_{\hat{\mathcal{D}}_u^j}(A)|$$

$$\leq 2 \sum_{i\in\mathcal{T}} \zeta_i \sum_{j\in\mathcal{T}} \zeta_j \sup_{A\in\mathcal{A}_{\mathcal{H}\triangle\mathcal{H}}} |\Pr_{\hat{\mathcal{D}}_u^i}(A) - \Pr_{\hat{\mathcal{D}}_u^j}(A)|$$

$$\leq \sum_{i\in\mathcal{T}} \sum_{j\in\mathcal{T}} \zeta_i \zeta_j d_{\mathcal{H}\triangle\mathcal{H}}(\hat{\mathcal{D}}_u^i, \hat{\mathcal{D}}_u^j). \qquad (19)$$

### A.3 PROOF ON LEMMA 6

**Lemma 6** *For any $\theta \in \Theta$, the domain divergence $d_{\mathcal{H}\triangle\mathcal{H}}(A,B)$ is bounded by the expectation of gradient divergence between domain $A$ and domain $B$.*

$$d_{\mathcal{H}\triangle\mathcal{H}}(A,B) \leq \frac{1}{\mu} d_{\mathcal{G}\circ\theta}(A,B), \qquad (20)$$

*where $d_{\mathcal{G}\circ\theta}(A,B)$ is the gradient divergence of model $\theta$ when training in two domains $A$ and $B$.*

*Proof.* By the definition of $d_{\mathcal{H} \triangle \mathcal{H}}(\cdot, \cdot)$ in (Arjovsky et al., 2020), we have:

$$d_{\mathcal{H} \triangle \mathcal{H}}(A, B) = 2 \sup_{\theta, \theta' \in \Theta} \left| \text{Pr}_{x \sim A}[f(x; \theta) \neq f(x; \theta')] - \text{Pr}_{x \sim B}[f(x; \theta) \neq f(x; \theta')] \right|, \qquad (21)$$

where $f(x; \theta)$ means the prediction function on data $x$ with model parameter $\theta$. We chose $\theta'$ as parameter of the label function, then $f(x; \theta) \neq f(x; \theta')$ means the loss function $\mathcal{L}(x; \theta)$, so we have:

$$d_{\mathcal{H} \triangle \mathcal{H}}(A, B) = 2 \sup_{\theta \in \Theta} \left| \text{Pr}_{x \sim A}[\mathcal{L}(x; \theta)] - \text{Pr}_{x \sim B}[\mathcal{L}(x; \theta)] \right|$$

$$= 2 \sup_{\theta \in \Theta} |\mathcal{E}_A(\theta) - \mathcal{E}_B(\theta)|. \overset{(a)}{\leq} \frac{2}{\mu} \sup_{\theta \in \Theta} |\nabla \mathcal{E}_A(\theta) - \nabla \mathcal{E}_B(\theta)| \leq \frac{1}{\mu} d_{\mathcal{G} \circ \theta}(A, B). \quad (22)$$

Here, $d_{\mathcal{G} \circ \theta}(A, B)$ as the gradient divergence, given the model $\theta$ and $(a)$ holds due to Assumption 2.

### A.4 Proof on Theorem 1

From Lemma 3 (Zhang et al., 2023c), we have:

$$\mathcal{E}(\theta; \mathcal{D}_{\mathcal{Q}}) \leq \mathcal{E}(\theta; \mathcal{D}_{\mathcal{P}}) + \frac{1}{2} d_{\mathcal{H}\triangle\mathcal{H}}(\hat{\mathcal{D}}_{\mathcal{P}}, \mathcal{D}_{\mathcal{Q}}) + \zeta^*. \tag{23}$$

Here, we have $\mathcal{E}(\theta; \mathcal{D}_{\mathcal{P}}) = \sum_{u=1}^{U} \gamma_u \mathcal{E}(\theta; \mathcal{D}_u^i) = \sum_{i \in \mathcal{T}} \sum_{u=1}^{U} \gamma_u \mathcal{E}(\theta; \mathcal{D}_u^i)$, and

$$\mathcal{E}(\theta; \mathcal{D}_{\mathcal{Q}}) \leq \mathcal{E}(\theta; \mathcal{D}_{\mathcal{P}}) + \frac{1}{2} d_{\mathcal{H}\triangle\mathcal{H}}(\hat{\mathcal{D}}_{\mathcal{P}}, \mathcal{D}_{\mathcal{P}}) + \frac{1}{2} d_{\mathcal{H}\triangle\mathcal{H}}(\mathcal{D}_{\mathcal{P}}, \mathcal{D}_{\mathcal{Q}}) + \zeta^*$$

$$\leq \sum_{u \in \mathcal{U}} \gamma_u \left[ \mathcal{E}(\theta; \mathcal{D}_{\mathcal{P}}) + \frac{1}{2} d_{\mathcal{H}\triangle\mathcal{H}}(\hat{\mathcal{D}}_{\mathcal{P}}, \mathcal{D}_{\mathcal{P}}) + \frac{1}{2} d_{\mathcal{H}\triangle\mathcal{H}}(\mathcal{D}_{\mathcal{P}}, \mathcal{D}_{\mathcal{Q}}) \right] + \zeta^*$$

$$\leq \sum_{u \in \mathcal{U}} \gamma_u \left[ \mathcal{E}(\theta; \hat{\mathcal{D}}_u) + \frac{1}{2} d_{\mathcal{H}\triangle\mathcal{H}}(\hat{\mathcal{D}}_u, \mathcal{D}_{\mathcal{P}}) + \frac{1}{2} d_{\mathcal{H}\triangle\mathcal{H}}(\mathcal{D}_{\mathcal{P}}, \mathcal{D}_{\mathcal{Q}}) + \sqrt{\frac{\log M + \log \frac{1}{\delta}}{2N_u}} \right] + \zeta^*$$

$$\overset{(a)}{\leq} \sum_{u \in \mathcal{U}} \gamma_u \left[ \mathcal{E}(\theta; \hat{\mathcal{D}}_u) + \sum_{v \in \mathcal{U}} \frac{1}{2} d_{\mathcal{H}\triangle\mathcal{H}}(\hat{\mathcal{D}}_u, \hat{\mathcal{D}}_v) + \frac{1}{2} d_{\mathcal{H}\triangle\mathcal{H}}(\mathcal{D}_{\mathcal{P}}, \mathcal{D}_{\mathcal{Q}}) \right.$$

$$\left. + \sqrt{\frac{\log M + \log \frac{1}{\delta}}{2N_u}} \right] + \zeta^*$$

$$\leq \sum_{t \in \mathcal{T}} \sum_{u \in \mathcal{U}} \gamma_u \left[ \mathcal{E}(\theta; \hat{\mathcal{D}}_u^t) + \sum_{v \in \mathcal{U}} \frac{1}{2} d_{\mathcal{H}\triangle\mathcal{H}}(\hat{\mathcal{D}}_u^t, \hat{\mathcal{D}}_v) + \frac{1}{2} d_{\mathcal{H}\triangle\mathcal{H}}(\mathcal{D}_{\mathcal{P}}, \mathcal{D}_{\mathcal{Q}}) \right.$$

$$\left. + \sqrt{\frac{\log M + \log \frac{1}{\delta}}{2N_u}} \right] + \zeta^*$$

$$\overset{(b)}{\leq} \sum_{t \in \mathcal{T}} \sum_{u \in \mathcal{U}} \gamma_u \left[ \mathcal{E}(\theta; \hat{\mathcal{D}}_u^t) + \sum_{v \in \mathcal{U}} \sum_{j \in \mathcal{T}} \frac{1}{2} d_{\mathcal{H}\triangle\mathcal{H}}(\hat{\mathcal{D}}_u^t, \hat{\mathcal{D}}_v^j) + \frac{1}{2} d_{\mathcal{H}\triangle\mathcal{H}}(\mathcal{D}_{\mathcal{P}}, \mathcal{D}_{\mathcal{Q}}) \right.$$

$$\left. + \sqrt{\frac{\log M + \log \frac{1}{\delta}}{2N_u}} \right] + \zeta^*$$

$$\overset{(c)}{\leq} \sum_{t \in \mathcal{T}} \sum_{u \in \mathcal{U}} \gamma_u \left[ \mathcal{E}(\theta; \hat{\mathcal{D}}_u^t) + \sum_{v \in \mathcal{U}} \sum_{j \in \mathcal{T}} \frac{1}{2} d_{\mathcal{H}\triangle\mathcal{H}}(\hat{\mathcal{D}}_u^t, \hat{\mathcal{D}}_u^j) + \sum_{v \in \mathcal{U}} \sum_{j \in \mathcal{T}} \frac{1}{2} d_{\mathcal{H}\triangle\mathcal{H}}(\hat{\mathcal{D}}_u^t, \hat{\mathcal{D}}_v^t) \right.$$

$$\left. + \frac{1}{2} d_{\mathcal{H}\triangle\mathcal{H}}(\mathcal{D}_{\mathcal{P}}, \mathcal{D}_{\mathcal{Q}}) + \sqrt{\frac{\log M + \log \frac{1}{\delta}}{2N_u}} \right] + \zeta^*, \tag{24}$$

where $(a)$ is according to Lemma 4, $(b)$ is according to Lemma 5, $(c)$ is according to Lemma 1. Simplify Eq. 24, we have

$$\mathcal{E}(\theta; \mathcal{D}_{\mathcal{Q}}) \leq \sum_{i \in \mathcal{T}} \sum_{u \in \mathcal{U}} \gamma_u \left[ \mathcal{E}(\theta; \mathcal{D}_u^i) + \sum_{j \in \mathcal{T}} d_{\mathcal{H}\triangle\mathcal{H}}(\mathcal{D}_u^i, \mathcal{D}_u^j) + \sum_{v \in \mathcal{U}} d_{\mathcal{H}\triangle\mathcal{H}}(\mathcal{D}_u^i, \mathcal{D}_v^i) \right.$$

$$\left. + \sqrt{\frac{\log M + \log \frac{1}{\delta}}{2N_u}} \right] + \zeta^*. \tag{25}$$

### A.5 Proof on Theorem 2

Let $\hat{\mathcal{D}}_u$ be the sampled counterpart from the domain $\mathcal{D}_u$, we have $\mathcal{E}_{\hat{\mathcal{D}}_u}$ is an empirical risk of $\mathcal{D}_u$, i.e., $\mathcal{E}_{\hat{\mathcal{D}}_u} = 1/N_u \sum_{i=1}^{N_u} \mathcal{L}(f(x_u^i; \theta), y_u^i)$. We also have expected risk $\mathcal{E}_{\mathcal{D}_u}$ defined as $\mathcal{E}_{\mathcal{D}_u} = \mathbb{E}_{(x, y \in \mathcal{D}_u)}[\mathcal{L}(f(x; \theta), y)]$. For a given $\theta \in \Theta$, with the definition of generalization bound, the

following inequality holds with at most $\frac{\delta}{U}$ for each domain $\hat{D}_u$ ($U$ is the number of users, which is also the number of spatial tasks).

$$\mathcal{E}_{\hat{\mathcal{D}}_u}(\theta) - \mathcal{E}_{\mathcal{D}_u}(\theta) > \sqrt{\frac{\log M + \log U/\delta}{2N_u}}. \tag{26}$$

Moreover, from Lemma 3, we have $|\mathcal{E}_{\mathcal{D}_u}(\theta) - \mathcal{E}_{\mathcal{D}_\mathcal{Q}}(\theta)| \leq \frac{1}{2}d_{\mathcal{H}\triangle\mathcal{H}}(\mathcal{D}_u, \mathcal{D}_\mathcal{Q})$ for each user $u$, and $|\mathcal{E}_{\hat{\mathcal{D}}_u^t}(\theta) - \mathcal{E}_{\hat{\mathcal{D}}_u}(\theta)| \leq \frac{1}{2}d_{\mathcal{H}\triangle\mathcal{H}}(\hat{\mathcal{D}}_u^t, \hat{\mathcal{D}}_u)$ for each temporal task $t$. Then let us consider (26), we can obtain the following inequalities with the probability at least greater than $1 - \frac{\delta}{U}$:

$$\min_{\theta'} \mathcal{E}_{\hat{\mathcal{D}}_u^t}(\theta') \leq \mathcal{E}_{\hat{\mathcal{D}}_u^t}(\theta) \leq \mathcal{E}_{\hat{\mathcal{D}}_u}(\theta) + \frac{1}{2}d_{\mathcal{H}\triangle\mathcal{H}}(\hat{\mathcal{D}}_u^t, \hat{\mathcal{D}}_u)$$

$$\leq \mathcal{E}_{\mathcal{D}_u}(\theta) + \frac{1}{2}d_{\mathcal{H}\triangle\mathcal{H}}(\hat{\mathcal{D}}_u^t, \hat{\mathcal{D}}_u) + \sqrt{\frac{\log M + \log U/\delta}{2N_u}}$$

$$\leq \mathcal{E}_{\mathcal{D}_\mathcal{Q}}(\theta) + \frac{1}{2}d_{\mathcal{H}\triangle\mathcal{H}}(\hat{\mathcal{D}}_u^t, \hat{\mathcal{D}}_u) + \frac{1}{2}d_{\mathcal{H}\triangle\mathcal{H}}(\hat{\mathcal{D}}_u, \mathcal{D}_\mathcal{Q}) + \sqrt{\frac{\log M + \log U/\delta}{2N_u}}. \tag{27}$$

We denote the local optimal on each client of source set $u$, $u \in \mathcal{U}$ as $\theta_u^*$. If we choose a specific parameter $\theta_\mathcal{T}^* = \min_\theta \mathcal{E}_{\mathcal{D}_\mathcal{Q}}(\theta)$ which is the local optimal on the unseen domain $\mathcal{T}$, the above third inequality still holds. Then, we can rewrite the above inequalities into:

$$\mathcal{E}_{\hat{\mathcal{D}}_u^t}(\theta_u^*) \leq \mathcal{E}_{\mathcal{D}_\mathcal{Q}}(\theta_u^*) + \frac{1}{2}d_{\mathcal{H}\triangle\mathcal{H}}(\hat{\mathcal{D}}_u^t, \hat{\mathcal{D}}_u) + \frac{1}{2}d_{\mathcal{H}\triangle\mathcal{H}}(\hat{\mathcal{D}}_u, \mathcal{D}_\mathcal{Q}) + \sqrt{\frac{\log M + \log U/\delta}{2N_u}}. \tag{28}$$

Considering on each domain, equation (28) holds. By a similar derivation process, we can obtain the inequality between $\mathcal{T}$ and $\hat{\mathcal{D}}$ with the probability at least greater than $1 - \delta$.

$$\sum_{t \in \mathcal{T}} \sum_{u \in \mathcal{U}} \gamma_u \zeta_t \mathcal{E}_{\hat{\mathcal{D}}_u^t}(\theta_u^*) \leq \mathcal{E}_{\mathcal{D}_\mathcal{Q}}(\theta_u^*) \tag{29}$$

$$+ \sum_{t \in \mathcal{T}} \sum_{u \in \mathcal{U}} \gamma_u \zeta_t \left[ \frac{1}{2}d_{\mathcal{H}\triangle\mathcal{H}}(\hat{\mathcal{D}}_u, \mathcal{D}_\mathcal{Q}) + \frac{1}{2}d_{\mathcal{H}\triangle\mathcal{H}}(\hat{\mathcal{D}}_u^t, \hat{\mathcal{D}}_u) + \sqrt{\frac{\log M + \log U/\delta}{2N_u}} \right].$$

From the above equation, we have Theorem 2 with the global model $\theta$ after R rounds FL. For instance,

$$\mathcal{E}_{\mathcal{D}_\mathcal{Q}}(\theta^R) - \mathcal{E}_{\mathcal{D}_\mathcal{Q}}(\theta_{\mathcal{D}_\mathcal{Q}}^*)$$

$$\leq \sum_{t \in \mathcal{T}} \sum_{u \in \mathcal{U}} \gamma_u \zeta_t \left[ \mathcal{E}_{\hat{\mathcal{D}}_u^t}(\theta) - \mathcal{E}_{\hat{\mathcal{D}}_u^t}(\theta_u^*) + d_{\mathcal{H}\triangle\mathcal{H}}(\hat{\mathcal{D}}_u, \mathcal{D}_\mathcal{Q}) + d_{\mathcal{H}\triangle\mathcal{H}}(\hat{\mathcal{D}}_u^t, \hat{\mathcal{D}}_u) \right.$$

$$\left. + \frac{\sqrt{\log M + \log \frac{1}{\delta}}}{\sqrt{2N_u}} + \frac{\sqrt{\log M + \log \frac{U}{\delta}}}{\sqrt{2N_u}} \right] + \zeta^* \tag{30}$$

$$\leq \sum_{t \in \mathcal{T}} \sum_{u \in \mathcal{U}} \gamma_u \zeta_t \left[ \mathcal{E}_{\hat{\mathcal{D}}_u^t}(\theta) + d_{\mathcal{H}\triangle\mathcal{H}}(\hat{\mathcal{D}}_u, \mathcal{D}_\mathcal{Q}) + d_{\mathcal{H}\triangle\mathcal{H}}(\hat{\mathcal{D}}_u^t, \hat{\mathcal{D}}_u) + \frac{\sqrt{\log \frac{M}{\delta}} + \sqrt{\log \frac{UM}{\delta}}}{\sqrt{2N_u}} \right] + \zeta^*.$$

To further analyze the convergence bound, we consider the Assumption 3. For instance,

$$\mathcal{E}_{\mathcal{D}_{\mathcal{Q}}}(\theta^R) - \mathcal{E}_{\mathcal{D}_{\mathcal{Q}}}(\theta^*_{\mathcal{D}_{\mathcal{Q}}})$$

$$\leq \sum_{t \in \mathcal{T}} \sum_{u \in \mathcal{U}} \gamma_u \zeta_t \left[ \mathcal{E}_{\hat{\mathcal{D}}_u}(\theta) + d_{\mathcal{H}\triangle\mathcal{H}}(\hat{\mathcal{D}}_u, \mathcal{D}_{\mathcal{Q}}) + d_{\mathcal{H}\triangle\mathcal{H}}(\hat{\mathcal{D}}_u^t, \hat{\mathcal{D}}_u) + \frac{\sqrt{\log \frac{M}{\delta}} + \sqrt{\log \frac{UM}{\delta}}}{\sqrt{2N_u}} \right] + \zeta^* \tag{31}$$

$$\leq \sum_{t \in \mathcal{T}} \sum_{u \in \mathcal{U}} \gamma_u \zeta_t \left[ \mathcal{E}_{\hat{\mathcal{D}}_u}(\theta) + d_{\mathcal{H}\triangle\mathcal{H}}(\hat{\mathcal{D}}_u^t, \hat{\mathcal{D}}_u) + d_{\mathcal{H}\triangle\mathcal{H}}(\hat{\mathcal{D}}_u, \mathcal{D}_{\mathcal{P}}) + d_{\mathcal{H}\triangle\mathcal{H}}(\mathcal{D}_{\mathcal{P}}, \mathcal{D}_{\mathcal{Q}}) \right. \tag{32}$$

$$\left. + \frac{\sqrt{\log \frac{M}{\delta}} + \sqrt{\log \frac{UM}{\delta}}}{\sqrt{2N_u}} \right] + \zeta^*$$

$$\overset{(b)}{\leq} \sum_{t \in \mathcal{T}} \sum_{u \in \mathcal{U}} \gamma_u \zeta_t \left[ \mathcal{E}_{\hat{\mathcal{D}}_u}(\theta) + d_{\mathcal{H}\triangle\mathcal{H}}(\hat{\mathcal{D}}_u^t, \hat{\mathcal{D}}_u) + \sum_{v \in \mathcal{U}} \frac{d_{\mathcal{H}\triangle\mathcal{H}}(\hat{\mathcal{D}}_u, \hat{\mathcal{D}}_v)}{\mu} + d_{\mathcal{H}\triangle\mathcal{H}}(\mathcal{D}_{\mathcal{P}}, \mathcal{D}_{\mathcal{Q}}) \right. \tag{33}$$

$$\left. + \frac{\sqrt{\log \frac{M}{\delta}} + \sqrt{\log \frac{UM}{\delta}}}{\sqrt{2N_u}} \right] + \zeta^*.$$

$$\overset{(c)}{\leq} \sum_{t \in \mathcal{T}} \sum_{u \in \mathcal{U}} \gamma_u \zeta_t \left[ \mathcal{E}_{\hat{\mathcal{D}}_u}(\theta) + \sum_{j \in \mathcal{T}} \frac{d_{\mathcal{H}\triangle\mathcal{H}}(\hat{\mathcal{D}}_u^t, \hat{\mathcal{D}}_u^j)}{\mu} + \sum_{v \in \mathcal{U}} \frac{d_{\mathcal{H}\triangle\mathcal{H}}(\hat{\mathcal{D}}_u^t, \hat{\mathcal{D}}_v^t)}{\mu} + d_{\mathcal{H}\triangle\mathcal{H}}(\mathcal{D}_{\mathcal{P}}, \mathcal{D}_{\mathcal{Q}}) \right. \tag{34}$$

$$\left. + \frac{\sqrt{\log \frac{M}{\delta}} + \sqrt{\log \frac{UM}{\delta}}}{\sqrt{2N_u}} \right] + \zeta^*.$$

holds due to Lemma 5. Applying Lemma 6, we have:We have $(b)$ holds due to Lemma **??** and $(c)$ holds due to Lemma 5. Applying Lemma 6, we have:

$$\mathcal{E}_{\mathcal{D}_{\mathcal{Q}}}(\theta^R) - \mathcal{E}_{\mathcal{D}_{\mathcal{Q}}}(\theta^*_{\mathcal{D}_{\mathcal{Q}}})$$

$$\leq \sum_{t \in \mathcal{T}} \sum_{u \in \mathcal{U}} \gamma_u \zeta_t \left[ \mathcal{E}_{\hat{\mathcal{D}}_u}(\theta) + \sum_{j \in \mathcal{T}} \frac{d_{\mathcal{G}\circ\theta}(\hat{\mathcal{D}}_u^t, \hat{\mathcal{D}}_u^j)}{\mu} + \sum_{v \in \mathcal{U}} \frac{d_{\mathcal{G}\circ\theta}(\hat{\mathcal{D}}_u^t, \hat{\mathcal{D}}_v^t)}{\mu} + d_{\mathcal{H}\triangle\mathcal{H}}(\mathcal{D}_{\mathcal{P}}, \mathcal{D}_{\mathcal{Q}}) \right. \tag{35}$$

$$\left. + \frac{\sqrt{\log \frac{M}{\delta}} + \sqrt{\log \frac{UM}{\delta}}}{\sqrt{2N_u}} \right] + \zeta^*.$$

## B  GRADIENT ALIGNMENT UPDATE RULE

We consider the parameter update rule $\theta^{(\tau,r+1)} = \theta^{(\tau,r)} - \eta x$, where $\eta$ denotes the learning rate and $x$ is the update direction to be determined. Our goal is to select $x$ such that not only the average loss $\bar{g}^{(r)}$ decreases, but each individual task loss decreases as well. To enforce this, we consider the worst generalization case among all seen tasks. Specifically,

$$\texttt{GAP}(\theta, x) = \max_{t \in \mathcal{T}} \left\{ \frac{1}{\eta} \mathcal{L}(\theta^{(\tau,r)} - \eta x; \mathcal{D}^t) - \mathcal{L}(\theta^{(\tau,r)}; \mathcal{D}^t) \right\} \approx \min_{t \in \mathcal{T}} \langle g^{(t,r)}, x \rangle. \tag{36}$$

Here, we use $g^{(t,r)}$ to denote, for simplicity, the gradient of the model at the current task $\tau$ when trained on the dataset of task $t$. Under the spatio gradient alignment setting, the spatio task is handled by taking $g_u^{(t,r)}$ as the gradient from client $u$, and the aggregation is performed over the set of $U$ clients rather than over the set of $T$ tasks.

To derive the invariant update direction $g_G$, we treat $x = g_G$ as the optimization variable and formulate the following maximization problem. Let $\phi = \kappa^2 \|\bar{g}^{(r)}\|^2$. The Lagrangian becomes

$$\max_{x} \min_{\lambda, \gamma} \left( \sum_{t \in \mathcal{T}} \gamma_t g^{(t,r)} \right)^\top x - \frac{\lambda}{2} \|\bar{g}^{(r)} - x\|^2 + \frac{\lambda}{2} \phi, \quad \text{s.t. } \lambda \geq 0. \tag{37}$$

Because the formulation is convex and satisfies Slater's condition for $\kappa > 0$ (and trivially holds for $\kappa = 0$), strong duality applies. Hence, we can exchange the $\min$ and $\max$ operators:

$$\min_{\lambda, \gamma} \max_{x} \underbrace{\left( \sum_{t \in \mathcal{T}} \gamma_t g^{(t,r)} \right)^\top x - \frac{\lambda}{2} \|\bar{g}^{(r)} - x\|^2 + \frac{\lambda}{2} \phi}_{A_1}, \quad \text{s.t. } \lambda \geq 0. \tag{38}$$

Fixing $(\lambda, \gamma)$ and optimizing over $x$, the optimality condition $\partial A_1 / \partial x = 0$ yields

$$\lambda(x - \bar{g}^{(r)}) - \sum_{t=1}^{T} \gamma_t g^{(t,r)} = 0,$$

which implies

$$x = \bar{g}^{(r)} + \left( \sum_{t=1}^{T} \gamma_t g^{(t,r)} \right) / \lambda. \tag{39}$$

Therefore, we have the followings:

$$A_1 = \left( \sum_{t=1}^{T} \gamma_t g^{(t,r)} \right)^\top \left( \bar{g}^{(r)} + \left( \sum_{t=1}^{T} \gamma_t g^{(t,r)} \right) / \lambda \right) - \frac{\lambda}{2} \| \bar{g}^{(r)} - \left( \bar{g}^{(r)} + \left( \sum_{t=1}^{T} \gamma_t g^{(t,r)} \right) / \lambda \right) \|^2 + \frac{\lambda}{2} \phi$$

$$= \left( \sum_{t=1}^{T} \gamma_t g^{(t,r)} \right)^\top \left( \bar{g}^{(r)} + \left( \sum_{t=1}^{T} \gamma_t g^{(t,r)} \right) / \lambda \right) - \frac{\lambda}{2} \| \frac{1}{\lambda} \sum_{t=1}^{T} \gamma_t g^{(t,r)} \|^2 + \frac{\lambda}{2} \phi. \tag{40}$$

Substituting the shorthand $g_\Gamma^{(r)} = \sum_{t=1}^{T} \gamma_t g^{(t,r)}$ into equation 38, we obtain

$$A_1 = g_\Gamma^{(r)\top} \left( \bar{g}^{(r)} + g_\Gamma^{(r)} / \lambda \right) - \frac{\lambda}{2} \| g_\Gamma^{(r)} / \lambda \|^2 + \frac{\lambda}{2} \phi$$

$$= g_\Gamma^{(r)\top} \bar{g}^{(r)} + \frac{1}{\lambda} g_\Gamma^{(r)\top} g_\Gamma^{(r)} - \frac{1}{2\lambda} \| g_\Gamma^{(r)} \|^2 + \frac{\lambda}{2} \phi$$

$$= g_\Gamma^{(r)\top} \bar{g}^{(r)} + \frac{1}{2\lambda} \| g_\Gamma^{(r)} \|^2 + \frac{\lambda}{2} \phi. \tag{41}$$

Thus the problem in Eq. equation 38 reduces to

$$\min_{\lambda, \gamma} \underbrace{g_\Gamma^{(r)\top} \bar{g}^{(r)} + \frac{1}{2\lambda} \| g_\Gamma^{(r)} \|^2 + \frac{\lambda}{2} \phi}_{A_2}. \tag{42}$$

To obtain the optimal $\lambda$, we differentiate $A_2$:

$$\frac{\partial}{\partial\lambda}A_2 = -\frac{1}{2\lambda^2}\|g_\Gamma^{(r)}\|^2 + \frac{1}{2}\phi = 0,$$

which gives

$$\lambda = \|g_\Gamma^{(r)}\|/\phi^{1/2}.$$

Finally, inserting this expression back into equation 42 and using equation 39, we obtain the invariant gradient direction:

$$g_G = \bar{g}^{(r)} + \frac{\kappa\|\bar{g}^{(r)}\|}{\|g_{\Gamma^*}^{(r)}\|}g_{\Gamma^*}^{(r)} \quad \text{s.t.} \quad \Gamma^* = \arg\min_\Gamma \Gamma\mathbf{g}^{(r)} \cdot \bar{g}^{(r)} + \kappa\|\bar{g}^{(r)}\|\|g_\Gamma^{(r)}\|. \tag{43}$$

This concludes the derivation.

## C  DETAILED ALGORITHMS

**Algorithm 1:** The box refers to the **S**patio gr**A**dient **M**atching (SAM), the box refers to the **T**emporal gr**A**dient **M**atching (TAM), the box refers to the **p**rototypical **c**oreset **s**election (PCS).

**Input:** set of source clients $\mathcal{U}$, number of communication rounds $R$, local learning rate $\eta$, global learning rate $\eta_g$, searching space hyper-parameter $\kappa$.

**Output:** $\theta_g^{(R)}$

1 **Clients Update:**

2 **for** *client* $u \in \mathcal{U}$ **do**

3     **Receive** global model $\theta_u^{(r)} = \theta_g^{(r)}$;

4     Compute $p^l = \frac{1}{\sum_{t=1}^{T}|\mathcal{N}_l^t|}\left[ g(\widetilde{x}^l;\phi) \cdot \sum_{j=1}^{t-1}|\mathcal{N}_l^j| + \sum_{i \in \mathcal{N}_l^t} g(x_i;\phi) \right] \cdot \mathbb{1}\{y_j = l\}$,

5     Initialize learnable coefficient set $A = \{a_i | i \in \mathcal{N}_l^t\}$

6     Solve $\widetilde{X}^l = \arg\min_A \left\| \left[ \frac{1}{|\mathcal{M}^l|}\sum_{i \in \mathcal{M}^l} g(x_i;\phi) + \frac{1}{|\mathcal{N}_l^t|}\sum_{i \in \mathcal{N}_l^t} a_i \cdot g(x_i;\phi) \right] - p^l \right\|^2$,

7     $\widetilde{x}^l = \text{MixStyle}(\widetilde{x}^l; x)$,

8     Save new proto into replay memory $\mathcal{M}^t = \widetilde{x}^l$.

9     **for** *local epoch* $e \in E$ **do**

10       Sample mini-batch $\zeta$ from local data $\mathcal{D}_u$;

11       Calculate gradient $g_u^{t,r,e} = \nabla\mathcal{E}(\theta_u^{(r,e)}, \zeta)$;

12     **end for**

13     Calculate $\widetilde{g}^t = \frac{1}{E}\sum_{e=1}^{E} g_u^{t,r,e}$.

14     **for** *task* $i = 1, \ldots, t-1$ **do**

15       Sample coreset $\zeta$ from replay memory $\mathcal{M}^i$ according to task $i$,

16       Calculate task-wise gradients: $\widetilde{g}_u^i = \nabla\mathcal{E}(\theta_u^{(r,e)}, \zeta)$.

17     **end for**

18     $\mathbf{g} = [\widetilde{g}_u^1, \ldots, \widetilde{g}_u^t]$, and $\bar{g} = \sum_{i=1}^{t} g_u^i$,

19     Solve: $\Gamma^* = \arg\min_\Gamma \Gamma\mathbf{g} \cdot \bar{g} + \kappa\|\bar{g}\|\|\Gamma\mathbf{g}^{(t,r)}\|$,

20     Update TAM: $g_{\text{TAM}} = \bar{g} + \frac{\kappa\|\bar{g}\|}{\|\Gamma^*\mathbf{g}^{(r)}\|}\Gamma^*\mathbf{g}^{(t,r)}$,

21     Model steps with aggregated gradient: $\theta_u^{(t,r)} = \theta_u^{(t,r-1)} - \eta_g g_{\text{TAM}}^{(t,r)}$.

22     Upload client's model $\theta_u^{(t,r+1)}$ to server;

23 **end for**

24 **Server Optimization:**

25 **for** *task* $t = 0, \ldots$ **do**

26     **for** *round* $r = 0, \ldots, R$ **do**

27       **Clients Updates;**

28       Calculate $g_u^{(t,r)} = \theta_u^{(t,r+1)} - \theta_u^{(t,r)}$, $\mathbf{g}^{(t,r)} = \{g_u^{(t,r)} | u \in \mathcal{U}\}$;

29       Calculate $g_{FL}^{(t,r)}$ (e.g., $g_{FL}^{(t,r)} = \frac{1}{U}\sum_{u=1}^{U} g_u^{(t,r)}$ as the FedAvg update);

30       Solve: $\Gamma^* = \arg\min_\Gamma \Gamma\mathbf{g}^{(t,r)} \cdot g_{\text{FL}}^{(t,r)} + \kappa\|g_{\text{FL}}^{(t,r)}\|\|\Gamma\mathbf{g}^{(t,r)}\|$,

31       Update SAM: $g_{\text{SAM}}^{(t,r)} = g_{\text{FL}}^{(t,r)} + \frac{\kappa\|g_{\text{FL}}^{(t,r)}\|}{\|\Gamma^*\mathbf{g}^{(t,r)}\|}\Gamma^*\mathbf{g}^{(t,r)}$,

32       Model steps with aggregated gradient: $\theta_u^{(t,r+1)} = \theta_u^{(t,r)} - \eta_u g_{\text{SAM}}^{(t,r)}$.

33     **end for**

34 **end for**

---

**Algorithm 2:** Prototypical Coreset Selection at task $t$

---

**Input:** Replay memory $\mathcal{M}$ with budget $|\mathcal{M}| = \sum_l^L |\mathcal{M}^l|$, new class data $\mathcal{N}_l^t$, encoder $\phi$

**Output:** updated replay memory $\mathcal{M}^l$

**1 for** *label* $l \in L$ **do**

**2**      **Step 1: Compute class prototype target**

**3**      $p^l = \frac{1}{\sum_{j=1}^{T} |\mathcal{N}_l^j|} \left[ g(\widetilde{x}^l; \phi) \cdot \sum_{j=1}^{t-1} |\mathcal{N}_l^j| + \sum_{i \in \mathcal{N}_l^t} g(x_i; \phi) \right] \cdot \mathbb{1}\{y_i = l\}.$

**4**      **Step 2: Initialize optimization variables**

**5**      Initialize coefficient set $A = \{a_i \mid i \in \mathcal{N}_l^t\} = \{1/\mathcal{N}_l^t \mid i \in \mathcal{N}_l^t\}.$

**6**      **Step 3: Solve prototype-matching objective**

**7**      **for** *epoch* $e \in E$ **do**

**8**          $\mathcal{L}_{\texttt{proto}} = \left\| \frac{1}{|\mathcal{M}^l|} \sum_{i \in \mathcal{M}^l} g(x_i; \phi) + \frac{1}{|\mathcal{N}_l^t|} \sum_{i \in \mathcal{N}_l^t} a_i \, g(x_i; \phi) - p^l \right\|^2.$

**9**          $A = A - \eta_A \nabla_A \mathcal{L}_{\texttt{proto}}.$

**10**      **end for**

**11**      $\widetilde{X}^l = \left\{ x_i \in \mathcal{N}_l^t \mid a_i \in \text{Top-}k(A) \right\}, \quad \text{s.t.} \quad k = |\mathcal{M}^l|.$

**12**      **Step 4: Style mixing if selected samples exceed memory**

**13**      **if** $|\widetilde{X}^l| > |\mathcal{M}^l|$ **then**

**14**          **foreach** $x \in \widetilde{X}^l$ **do**

**15**              $\widetilde{x}^l = \text{MixStyle}(\widetilde{x}^l; x)$ ;

**16**          **end foreach**

**17**      **end if**

**18**      **Step 5: Update replay memory**

**19**      $\mathcal{M}^l \leftarrow \widetilde{X}^l$

**20 end for**

---

# D RELATED WORKS

## D.1 IMPORTANCE-BASED SAMPLING

LGA (Dong et al., 2024) introduces a method to balance the contributions of different classes to the gradient, aiming to mitigate catastrophic forgetting caused by imbalance among incremental tasks. Re-Fed (Li et al., 2024b) presents a method for quantifying an importance score, which is utilized to selectively retain cached samples within the replay memory. FedWeIT (Yoon et al., 2021) partitions network weights into global federated and sparse task-specific parameters, enabling clients to selectively acquire knowledge through a weighted combination of others' task-specific parameters. FedSSI (Li et al., 2025c) introduces a regularization technique that estimates the importance of each synaptic weight change during training. It penalizes substantial changes to weights deemed important for previously learned tasks, thereby helping to preserve prior knowledge.

## D.2 PROTOTYPE-BASED LEARNING

SR-FDIL (Li et al., 2024c) introduces an approach that utilizes data from the local replay memory to train both the prototype generator and the discriminator on local devices. TagFed (Wang et al., 2024) proposes a method to identify repetitive data features from previous tasks and augment them for the current task prior to federation, thereby enhancing overall performance.

## D.3 GRADIENT MEMORY

GradMA (Luo et al., 2023) employs gradient projection on the client side, correcting gradients via quadrature optimization using stored gradients from other clients.

## D.4 GENERATIVE REPLAY MEMORY

FedCIL (Qi et al., 2023) introduces an efficient approach for training GAN-based replay memory in distributed systems. TARGET (Zhang et al., 2023b) introduces an approach that learns a server-side generative model capable of producing data that adheres to the global model distribution. This generated data is subsequently used to update the client-side student model via knowledge distillation. AF-FCL (Wuerkaixi et al., 2024) introduces a generative model that employs a learned normalizing flow to capture and retain the essential data distribution while effectively eliminating biased features. pFedDIL (Li et al., 2025d) proposes an approach that transfers knowledge across incremental tasks by using a small auxiliary classifier in each personalized model to distinguish its specific task from others. FBL (Dong et al., 2023) uses adaptive class-balanced pseudo labeling along with semantic compensation and relation consistency losses to generate reliable pseudo labels and balance gradient propagation, thereby mitigating the effects of background shifts.

## D.5 EPISODIC REPLAY MEMORY FOR CONTINUAL LEARNING

GEM (Lopez-Paz & Ranzato, 2017) introduced an episodic memory mechanism that stores a subset of data samples, enabling the estimation of task-specific gradients. This approach facilitates gradient projection, thereby mitigating catastrophic forgetting in CL. VR-MCL (Wu et al., 2024) introduced a meta CL approach that effectively utilizes data stored in the memory buffer.

Authors in (Qi et al., 2023) demonstrate that incorporating a GAN-based replay memory in a distributed system can be significantly affected by feature shifts among clients. To address this challenge, FedCIL introduces a distillation-based approach designed to mitigate discrepancies across different domains. GPM (Saha et al., 2021) introduces a method for storing gradient projections in replay memory as an alternative to retaining previous data, thereby facilitating CL. FS-DGPM (Deng et al., 2021) introduces an enhanced version of GPM, in which the projected gradients are flattened. This flattening process improves generalization and enhances robustness to noise caused by a sharp loss landscape.

# E  EXPERIMENTAL DETAILS

We utilize the pFLLib framework (Zhang et al., 2025) as FL core framework to design the FCL settings. All experiments are conducted using six NVIDIA GeForce RTX 4090 GPUs and two NVIDIA GeForce RTX 3090 GPUs. The detailed experimental configurations are outlined below:

## E.1  DATASETS

### E.1.1  HETEROGENEOUS FEDERATED CONTINUAL LEARNING SETTINGS

Our work investigates the behavior of various algorithms in a heterogeneous FCL setting. To align with a realistic and challenging non-IID federated scenario, we increase the difficulty by adopting the task design proposed by (Dohare et al., 2024), in which we construct a sequence of classification tasks by taking the classes in groups.

**Example 1** *For example, in case of binary classification, one task could involve differentiating chickens from llamas, while another might focus on differentiating phones from computers.*

To consider the performance of baselines under different level of heterogeneity, we consider two experimental scenarios. In the first, each task comprises 20 distinct classes. This setup represents the conventional task configuration commonly used in existing literature (Wuerkaixi et al., 2024). In the second, each task contains only 2 classes, creating a more challenging environment. In this case, models are more likely to overfit to individual tasks, making them more susceptible to catastrophic forgetting when adapting to new tasks. Simultaneously, client divergence becomes more pronounced under this configuration.

Specifically, we utilize two widely adopted benchmark datasets:

**Sequenced-CIFAR100.** The CIFAR100 dataset (Krizhevsky, 2009) consists of 100 object categories, with a total of 60,000 images. Each image has a resolution of $32 \times 32$ pixels. In case 1 task comprises 2 classes, we can form 4950 distinct tasks. In case 1 task comprises 20 classes, we can form more than $5 \times 10^{20}$ distinct tasks.

**Sequenced-ImageNet1K.** ImageNet1K dataset (Deng et al., 2009) contains 1,000 diverse object categories, with over 1.3 million high-resolution training images. All images are resized to $224 \times 224$ pixels during preprocessing. In case 1 task comprises 2 classes, we can form half a million tasks. We show the illustration for this case in Fig. 7. In case 1 task comprises 20 classes, we can form more than $3 \times 10^{41}$ distinct tasks. The scale and diversity of ImageNet1K pose greater challenges in terms of memory footprint, computational cost, and model scalability.

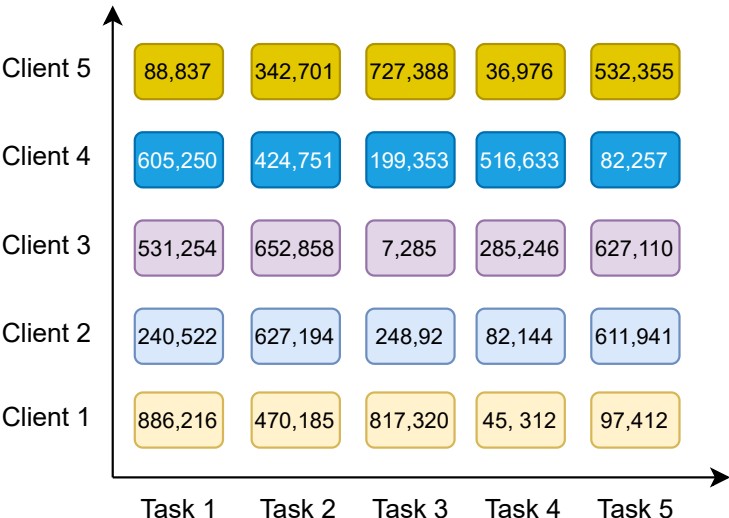

Figure 7: The data distribution when using S-ImageNet1K in case 1 task comprises 2 classes.

### E.2 BASELINES

We evaluate our approach against several established baselines from FL, and FCL. For conventional FL baselines, we compare with standard methods such as FedAvg (McMahan et al., 2017) , FedDBE (Zhang et al., 2023a), FedL2P (Lee et al., 2023), and FedAS (Yang et al., 2024), FedOMG (Nguyen et al., 2025). FedAvg serves as the foundational baseline in FL. FedL2P and FedAS focus on personalized FL, enabling models to adapt to client-specific tasks and thereby mitigating the effects of task heterogeneity. In contrast, FedDBE and FedOMG aim to construct a more robust global model by reducing inter-client bias, thereby enhancing generalization across both tasks and clients.

For FCL, we assess several state-of-the-arts, including FedWeIT (Yoon et al., 2021), GLFC (Dong et al., 2022), FedCIL (Qi et al., 2023), LANDER (Tran et al., 2024), TARGET (Zhang et al., 2023b), FedSSI (Li et al., 2025c), ReFed+ (Li et al., 2025a), and AF-FCL (Wuerkaixi et al., 2024). FedWeIT exemplifies approaches that allocate specialized expert modules for each task, allowing task-specific adaptation. GLFC uses a distillation-based approach to address catastrophic forgetting, considering both local and global aspects. FedCIL, LANDER, TARGET, and AF-FCL adopt generative replay strategies, training generative models on each client to synthesize pseudo-data for previously encountered tasks. Among these, AF-FCL is the most recent and directly addresses the challenges posed by heterogeneous FCL settings, making it a particularly relevant benchmark for comparison.

### E.3 EVALUATION METRICS

To evaluate the baselines, we utilize two standard metrics from the CL literature (Yoon et al., 2021), (Mirzadeh et al., 2021), which are well-suited for tracking the performance of a global model in FL, coined accuracy and averaged forgetting.

**Averaged Forgetting.** This metric measures the decline from a task's highest accuracy, which is typically achieved right after it is trained, to its final accuracy after all tasks have been learned. For $T$ tasks, the forgetting is defined as

$$AF = \frac{1}{T-1} \sum_{i=1}^{T-1} \max_{t \in [1:T-1]} (a_{t,i} - a_{T,i}).$$ (44)

As the model shifts focus to new tasks, its performance on earlier ones often decreases. Therefore, minimizing forgetting is important to maintain overall performance.

### E.4 ARCHITECTURE DETAILS

For CIFAR-10, CIFAR100, Digit10, and Office31, we adopt conventional ResNet-18 (He et al., 2016) as the backbone network architecture for all validation experiments. For S-ImageNet1K, we employ Swin Transformer Tiny (Swin-T) (Liu et al., 2021) as the backbone. It is noted that FCIL, LANDER, TARGET, FedL2P, FedWeIT and AF-FCL use addition generative networks or modify their network architectures, with details summarized in the following table. We denote FedWeIT (T) as the version theoretically proposed in the original paper, while FedWeIT (C) represents the configuration observed in our experimental implementation.

Specifically, FedWeIT augments the base model with sparse task-adaptive parameters, task-specific masks over local base parameters, and attention weights for inter-client knowledge transfer. FCIL, LANDER, and TARGET incorporate additional GANs to learn past task features. FedL2P introduces a meta-net that generates personalized hyper-parameters, such as batch normalization statistics and learning rates, adapted to each client's local data distribution to improve learning on non-IID data. AF-FCL additionally requires a normalizing flow generative model (NFlow[1]) for credibility estimation and generative replay mechanism, which guide selective retention and forgetting.

### E.5 TRAINING DETAILS

In our proposed heterogeneous federated continual learning framework for the S-CIFAR100 and S-ImageNet1K datasets, we consider a setting involving 10 clients with a client participation fraction

---

[1]NFlow refers to the normalizing flow model, where the example is provided in https://github.com/zaocan666/AF-FCL/blob/main/FLAlgorithms/PreciseFCLNet/model.py

Table 3: Architectural details of methods with modified models.

| Method | CIFAR-10, CIFAR100, Digit10, Office31 | | ImageNet1K | |
|--------|--------|--------|--------|--------|
| | Model | #Params | Model | #Params |
| FedAvg | ResNet-18 | 11.7 M | Swin-T | 28.8 M |
| FedSSI | ResNet-18 | 11.7 M | Swin-T | 28.8 M |
| ReFed+ | ResNet-18 | 11.7 M | Swin-T | 28.8 M |
| FCIL | ResNet-18 + GAN | 16.1 M | Swin-T + GAN | 49.7 M |
| LANDER | ResNet-18 + GAN | 16.1 M | Swin-T + GAN | 49.7 M |
| TARGET | ResNet-18 + GAN | 16.1 M | Swin-T + GAN | 49.7 M |
| FedL2P | ResNet-18 + Meta-Net | 13.5 M | Swin-T + Meta-Net | 32.6 M |
| FedWeIT (T) | Modified ResNet-18 | 596.2 M | Modified Swin-T | 7192.3 M |
| FedWeIT (C) | Modified LeNet | 171.8 B | | |
| AF-FCL | ResNet-18 + NFlow | 21.3 M | Swin-T + NFlow | 53.4 M |

of 1.0. We do not adopt a conventional non-IID distribution in this scenario; instead, each client is assigned distinct classes, which introduces a level of heterogeneity that is more challenging than typical non-IID configurations.

Additionally, we evaluate the proposed approach under non-IID conditions using four benchmark datasets: CIFAR-10, CIFAR100, Digit-10, and Office-31. For these experiments, we simulate data heterogeneity using the Dirichlet distribution with varying concentration parameters (e.g., $\alpha = 0.1$, 1.0, 10.0, and 100.0) to control the degree of non-IID-ness. The complete details of the experimental settings are provided in Table 4.

Table 4: Experimental Details. Settings for heterogeneous and non-IID distributed FCL.

| Attributes | Heterogeneous FCL | | Non-IID distributed FCL | | | |
|--------|--------|--------|--------|--------|--------|--------|
| | S-CIFAR100 | ImageNet1K | CIFAR10 | S-CIFAR100 | Digit10 | Office31 |
| Task size | 141 MB / 14 MB | 8 GB / 0.8 GB | 141 MB | 141 MB | 480 M | 88 M |
| Image number | 60K | 1.3M | 60K | 60K | 110K | 4.6K |
| Image Size | $3 \times 32 \times 32$ | $3 \times 224 \times 224$ | $3 \times 32 \times 32$ | $3 \times 32 \times 32$ | $1 \times 28 \times 28$ | $3 \times 300 \times 300$ |
| Task number | 5 / 50 | 50 / 500 | 5 | 10 | 4 | 3 |
| Batch Size | 128 | 128 | 64 | 64 | 64 | 32 |
| Learning Rate | 0.005 | 0.005 | 0.01 | 0.01 | 0.001 | 0.01 |
| Data heterogeneity | N/A | N/A | 0.1 | 10.0 | 0.1 | 1.0 |
| Client numbers | 10 | 10 | 10 | 10 | 10 | 10 |
| Local training epoch | 5 | 5 | 5 | 5 | 5 | 5 |
| Client selection ratio | 1.0 | 1.0 | 1.0 | 1.0 | 1.0 | 1.0 |
| Rounds per Task | 25 | 25 | 80 | 100 | 60 | 60 |

## F ADDITIONAL EXPERIMENTAL EVALUATIONS

### F.1 EXPERIMENTAL EVALUATIONS ON THE POPULAR CLASS DISTRIBUTION USED BY OTHER WORKS

The results in Table 5 show that when each task contains 20 classes, the problem becomes easier, leading to much lower forgetting across all methods compared to the 2-class setting. Even under this easier scenario, STAMP maintains a strong overall trade-off, achieving higher accuracy and competitive forgetting while keeping communication cost comparable to standard FL. At the same time, STAMP requires only modest GPU and disk resources, unlike methods such as LANDER or FedWeIT that consume significantly more memory. This efficiency highlights STAMP's robustness and practicality for real-world deployment, even when class distributions are less challenging.

Table 5: We report the average per-task performance of FCL under a setting where each task is assigned 20 classes. Evaluations are conducted using 10 clients (fraction = 1.0) across 5 independent trials. OOM refers to the out of memory in GPU. ↑ and ↓ indicate that higher and lower values are better, respectively. C→S and S→C denote communication from the client to the server and from the server to the client, respectively.

| Methods | Accuracy ↑ | AF ↓ | Avg. Comp. ↓ (Sec/Round) | Comm. Cost ↓ C→S | S → C | GPU (Peak) ↓ | Disk ↓ |
|---|---|---|---|---|---|---|---|
| **S-CIFAR100** ($U = 10, C = 20$) | | | | | | | |
| FedAvg | 27.2 (± 2.2) | 5.9 (± 0.9) | 27.6 sec | 44.6 MB | 44.6 MB | 1.92 GB | N/A |
| FedDBE | 28.3 (± 1.6) | 5.5 (± 0.7) | 28.3 sec | 44.6 MB | 44.6 MB | 1.91 GB | N/A |
| FedAS | 40.2 (± 1.1) | 30.7 (± 0.3) | 135.7 sec | 44.6 MB | 44.6 MB | 1.92 GB | N/A |
| FedOMG | 36.8 (± 1.4) | 8.5 (± 0.6) | 32.7 sec | 44.6 MB | 44.6 MB | 1.92 GB | N/A |
| GLFC | 29.8 (± 2.1) | 7.5 (± 0.4) | 167.8 sec | 88.2 MB | 46.5 MB | 3.83 GB | 22.1 MB |
| FedCIL | 32.4 (± 1.7) | 6.3 (± 1.2) | 199.3 sec | 95.3 MB | 44.6 MB | 4.21 GB | 18.5 MB |
| LANDER | 35.1 (± 1.3) | 5.4 (± 0.8) | 153.6 sec | 112.4 MB | 138.7 MB | 4.83 GB | 131.5 MB |
| TARGET | 32.1 (± 2.3) | 5.9 (± 1.6) | 236.4 sec | 112.4 MB | 44.6 MB | 3.65 GB | 18.5 MB |
| FedL2P | 30.2 (± 1.8) | 6.3 (± 1.3) | 78.1 sec | 56.3 MB | 56.3 MB | 2.56 GB | N/A |
| Re-Fed+ | 37.4 (± 1.6) | 6.3 (± 1.3) | 29.2 sec | 44.6 MB | 44.6 MB | 2.17 GB | 18.5 MB |
| FedWeIT | 37.3 (± 2.3) | 4.7 (± 0.8) | 38.7 sec | 44.2 MB | 44.2 MB | 7.21 GB | 6.1 GB |
| FedSSI | 39.2 (± 1.5) | 8.9 (± 1.1) | 61.7 sec | 44.6 MB | 44.6 MB | 2.53 GB | N/A |
| AF-FCL | 35.6 (± 0.4) | 5.2 (± 0.5) | 45.3 sec | 156.3 MB | 121.3 MB | 8.93 GB | N/A |
| **STAMP** | 41.3 (± 0.9) | 5.4 (± 0.6) | 56.3 sec | 44.6 MB | 44.6 MB | 1.92 GB | 16.3 MB |
| **S-ImageNet1K** ($U = 10, C = 20$) | | | | | | | |
| FedAvg | 17.3 (± 3.3) | 14.1 (± 0.2) | 1485.2 sec | 112.5 MB | 112.5 MB | 16.11 GB | N/A |
| FedDBE | 18.8 (± 5.2) | 13.9 (± 0.3) | 1572.7 sec | 112.5 MB | 112.5 MB | 16.11 GB | N/A |
| FedAS | 22.3 (± 5.0) | 18.2 (± 0.6) | 5108.5 sec | 112.5 MB | 112.5 MB | 16.11 GB | N/A |
| FedOMG | 21.2 (± 3.3) | 11.3 (± 0.7) | 1821.2 sec | 112.5 MB | 112.5 MB | 16.11 GB | N/A |
| GLFC | 22.5 (± 2.1) | 6.3 (± 0.2) | 5647.3 sec | 225.3 MB | 121.2 MB | 20.24 GB | 112.6 MB |
| FedCIL | 24.1 (± 2.8) | 7.3 (± 0.4) | 7120.3 sec | 245.5 MB | 112.5 MB | 23.47 GB | 184.3 MB |
| LANDER | 26.9 (± 1.4) | 7.8 (± 0.9) | 6825.8 sec | 267.4 MB | 453.6 MB | 26.54 GB | 1.31 GB |
| TARGET | 25.8 (± 3.8) | 6.7 (± 0.4) | 9958.2 sec | 287.4 MB | 112.5 MB | 21.08 GB | 184.3 MB |
| FedL2P | 22.3 (± 3.7) | 9.4 (± 0.6) | 3278.7 sec | 146.6 MB | 146.6 MB | 18.21 GB | N/A |
| Re-Fed+ | 25.4 (± 1.9) | 7.4 (± 0.6) | 1508.4 sec | 112.5 MB | 112.5 MB | 16.71 GB | 184.3 MB |
| FedWeIT | 24.8 (± 1.3) | 5.1 (± 0.8) | 1763.8 sec | 110.4 MB | 110.4 MB | 41.23 GB | 61.7 GB |
| FedSSI | 25.1 (± 2.4) | 8.6 (± 0.9) | 3111.8 sec | 287.4 MB | 112.5 MB | 17.66 GB | N/A |
| AF-FCL | 21.3 (± 5.1) | 4.5 (± 0.6) | 1823.7 sec | 421.3 MB | 336.8 MB | 46.81 GB | N/A |
| **STAMP** | 26.8 (± 2.3) | 5.8 (± 0.4) | 3041.2 sec | 112.5 MB | 112.5 MB | 16.11 GB | 152.6 MB |

## F.2 EXPERIMENTAL EVALUATIONS ON PRETRAINED MODELS

Figure 8 illustrates the performance of FedAvg and STAMP on the S-ImageNet1K dataset using a pretrained model. Given that the model is pretrained on the same dataset, the evaluation may suffer from overfitting. Consequently, the experimental results show no substantial performance difference between the two algorithms. Moreover, the issue of catastrophic forgetting appears to be minimal in this evaluation setting.

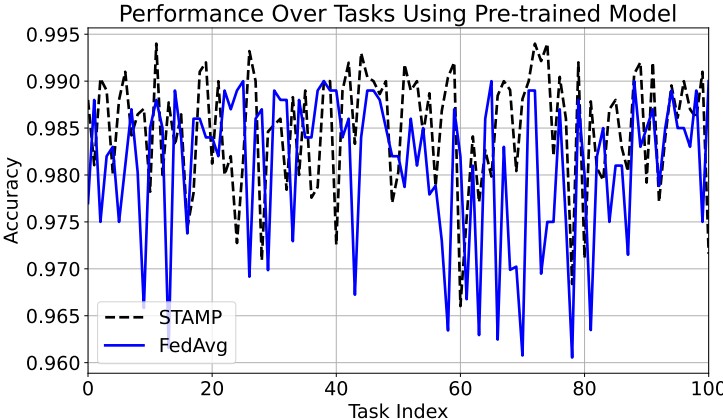

Figure 8: Accuracy on S-ImageNet1K with Pretrained Models.

### F.3 Experimental Evaluations on Catastrophic Forgetting

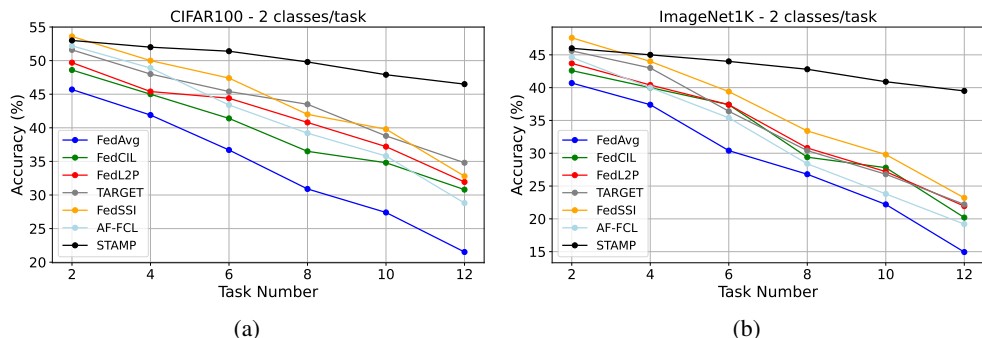

Figure 9: Analysis on forgetting curves.

Figure 9 shows that STAMP consistently exhibits substantially less performance degradation as the number of tasks increases, maintaining higher accuracy across both S-CIFAR100 and S-ImageNet1K. In contrast, other methods display similar downward trends, with accuracy declining more rapidly as tasks progress. Moreover, as illustrated in Figures 5 and 6, higher gradient angles between tasks correspond to more gradual decline in the forgetting curves, indicating less catastrophic forgetting.

### F.4 HYPER-PARAMETER TUNING FOR STAMP

In this section, we examine the impact of various hyperparameters through a series of experiments conducted on the ImageNet-1K dataset. For each experiment, one specific hyperparameter is varied while all other hyperparameters are held constant.

#### F.4.1 GRADIENT NORMALIZATION

Since STAMP is sensitive to the magnitude of local gradients, the presence of a dominant subset with disproportionately large gradient magnitudes can bias the optimization process toward that subset during gradient alignment. Figure 10 illustrates the impact of applying gradient normalization on both the client and server sides before performing gradient alignment. With gradient normalization in place, STAMP demonstrates a notable improvement in performance.

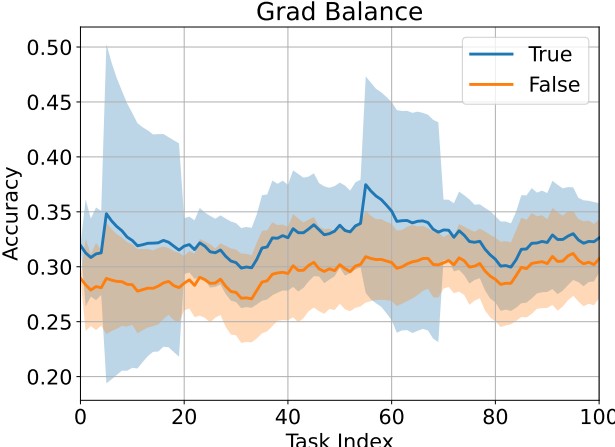

Figure 10: Analysis on Gradient Normalization.

#### F.4.2 GLOBAL TRAINING EPOCHS NUMBER PER ROUND

Fig 11 shows that using 25 training epochs achieves the best balance between performance and stability. Increasing the number of epochs beyond 25 does not lead to higher accuracy, while it results in increased forgetting, as indicated by the rise in average forgetting.

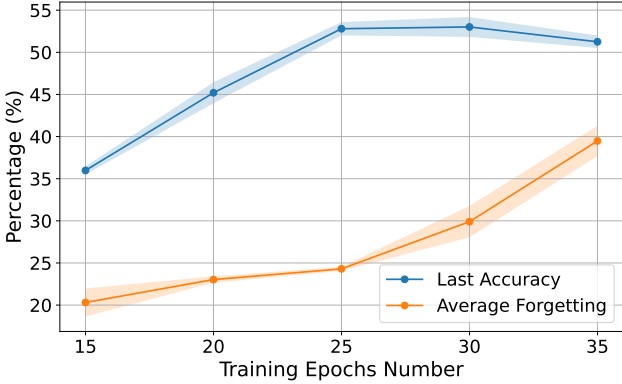

Figure 11: Analysis on Gradient Normalization.

### F.4.3 LOCAL EPOCH

Selecting the number of local epochs is crucial, as increasing the number of local epochs leads to a more accurate approximation of the local gradient trajectory. Figure 12 illustrates the performance of STAMP under varying numbers of local epochs.

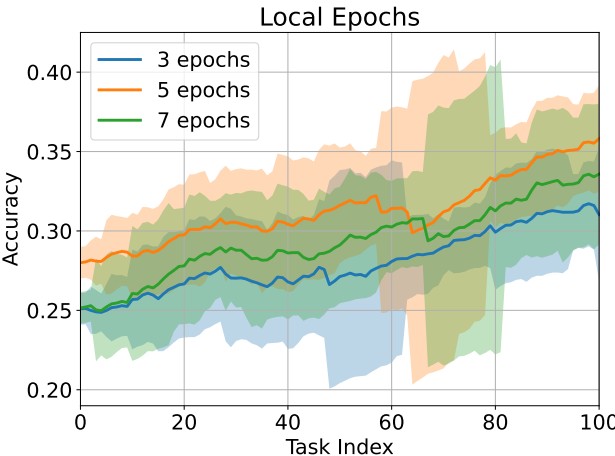

Figure 12: Analysis on different number of local epochs.

### F.4.4 LOCAL LEARNING RATE

Figure 13 illustrates the performance of STAMP under different local learning rate.

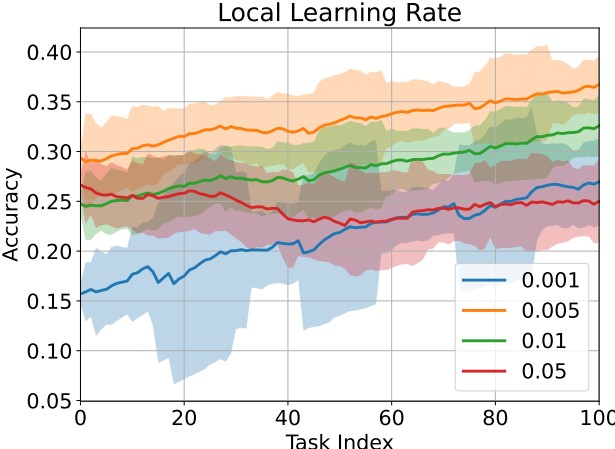

Figure 13: Analysis on different local learning rate.

### F.4.5 GRADIENT ALIGNMENT SEARCHING RADIUS

Figure 14 illustrates the impact of the search radius on gradient alignment in STAMP. Selecting an appropriate search radius (e.g., $0.5$) is critical for achieving an optimal gradient alignment solution. A smaller radius (e.g., $0.1$) constrains the search space too tightly, causing the solution to converge toward the average gradient and reducing matching effectiveness. Conversely, a larger radius (e.g., $0.75$) broadens the search space excessively, making it difficult to identify an optimal solution.

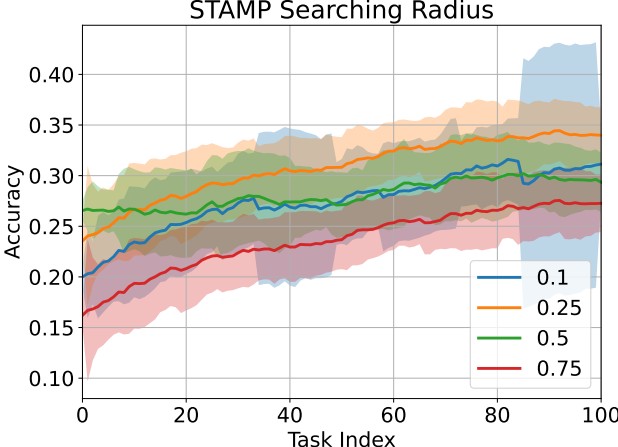

Figure 14: Analysis on different searching radius.

### F.4.6 GRADIENT ALIGNMENT STEP SIZE & MOMENTUM

Figures 15 and 16 demonstrate the effects of momentum and learning rate scheduling on gradient alignment performance.

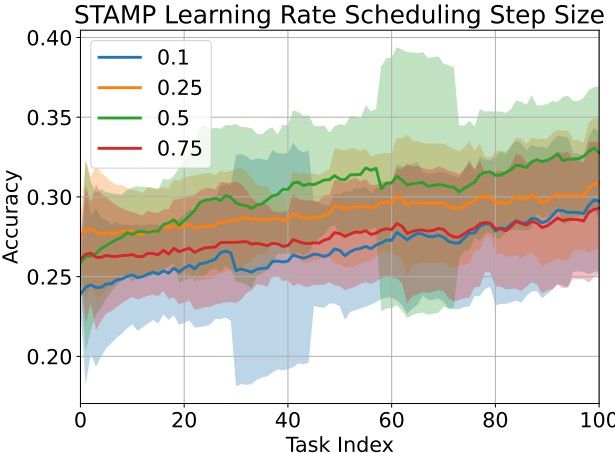

Figure 15: Analysis on different learning rate scheduling step size.

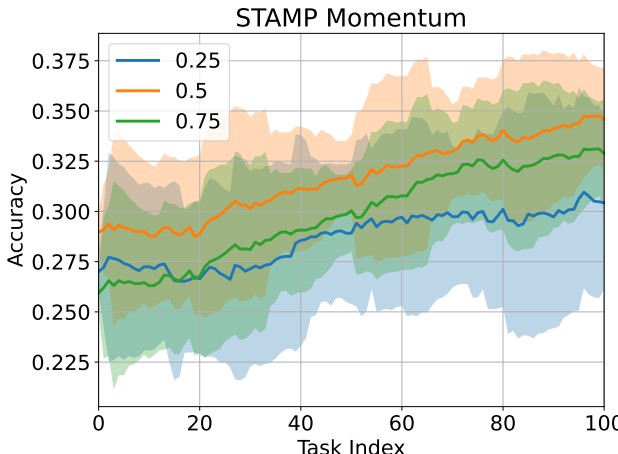

Figure 16: Analysis on different momentum for gradient alignment.

### F.4.7 GRADIENT ALIGNMENT NUMBER OF ROUNDS

Figure 17 illustrates the impact of the number of optimization steps on the efficiency of gradient alignment.

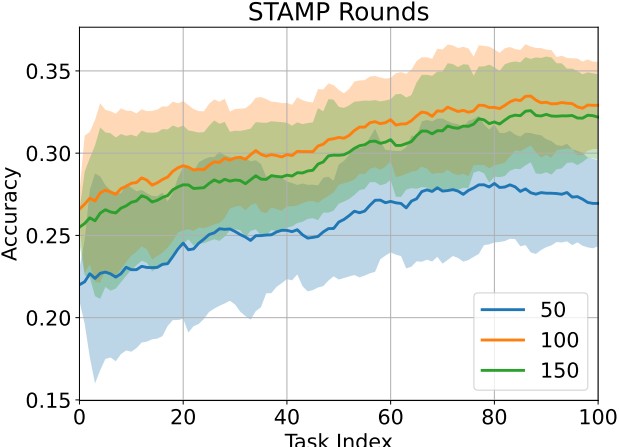

Figure 17: Analysis on different number of rounds

### F.4.8 GRADIENT ALIGNMENT SCHEDULING STEP SIZE

Figure 18 illustrates the performance of STAMP under various learning rate scheduler step sizes. Selecting an appropriate step size (e.g., 30) facilitates optimal gradient alignment decisions, thereby enhancing the stability and efficiency of FCL training.

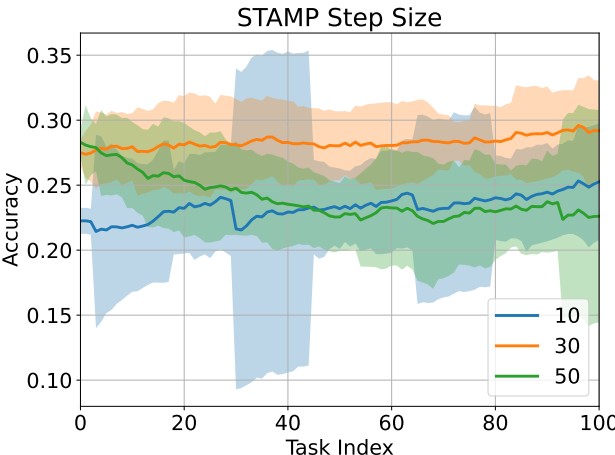

Figure 18: Analysis on different scheduling step size.

### F.4.9 GRADIENT ALIGNMENT LEARNING RATE

Figure 19 illustrates the effect of varying learning rates on the optimization of gradient alignment. The results indicate that STAMP achieves optimal performance when the learning rate is set to 25.

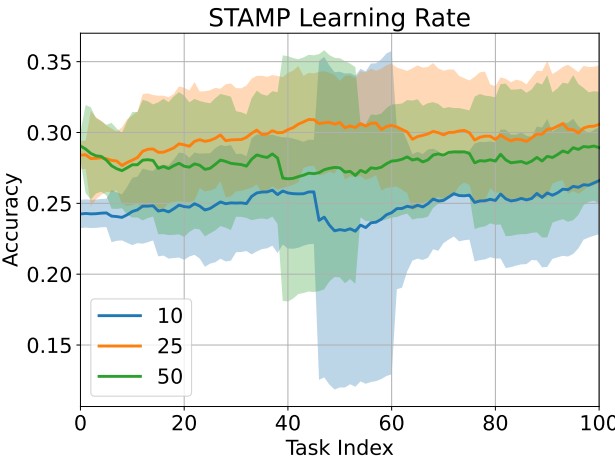

Figure 19: Analysis on different gradient alignment learning rate.

### F.4.10 GLOBAL UPDATE LEARNING RATE

The global update learning rate significantly influences the norm of the aggregated gradient. As shown in Figure 20a, selecting a lower learning rate can reduce the norm of the aggregated gradient (see Figure 20b). This reduction may lead to slower convergence or result in gradient magnitudes that are insufficient to escape sharp minima.

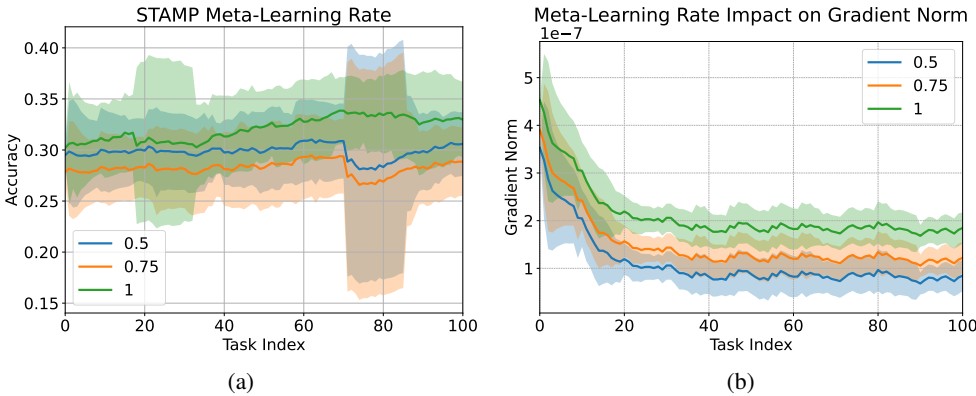

Figure 20: Analysis on global learning rate.

### F.5 EFFECTIVENESS OF GRADIENT ALIGNMENT

#### F.5.1 CASE OF 10 CLIENTS

To investigate the presence of gradient conflicts in federated learning (FL), we begin with a small-scale experiment involving 10 clients, each performing a classification task on the CIFAR-100 dataset, following the setup described above. We randomly select one client (denoted as client 1) and compute the cosine similarity between its gradient and those of the remaining 9 clients throughout the training process.

Figure 21 illustrates the cosine similarities between client 1 and each of the other clients (clients 2–10). It can be observed that under our proposed STAMP method, the gradients of client 1 are more consistently aligned with those of the other clients, as evidenced by higher cosine similarity values. This alignment indicates a reduction in gradient conflict and supports more stable collaborative learning.

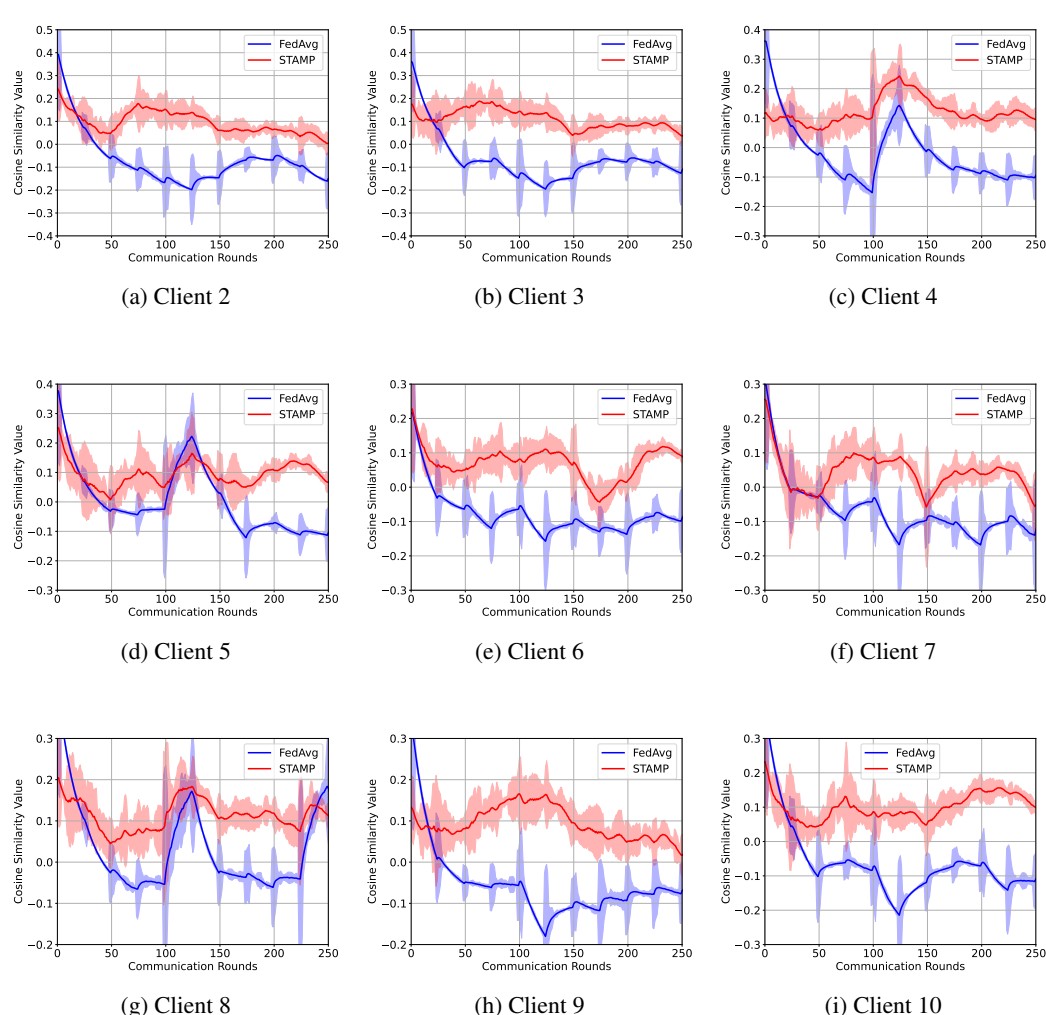

Figure 21: Cosine similarity between the gradient of client 1 and the gradients of clients 2–10. STAMP helps improve gradient alignment across clients by increasing cosine similarities.

To further quantify this effect, we aggregate the number of positive and negative cosine similarities across training rounds. As shown in Figure 22a and Figure 22b, the standard FedAvg method results in frequent gradient conflicts, indicated by a large number of negative similarities. In contrast, STAMP significantly reduces these conflicts, increasing the number of positively aligned gradients and thereby promoting more effective global model updates.

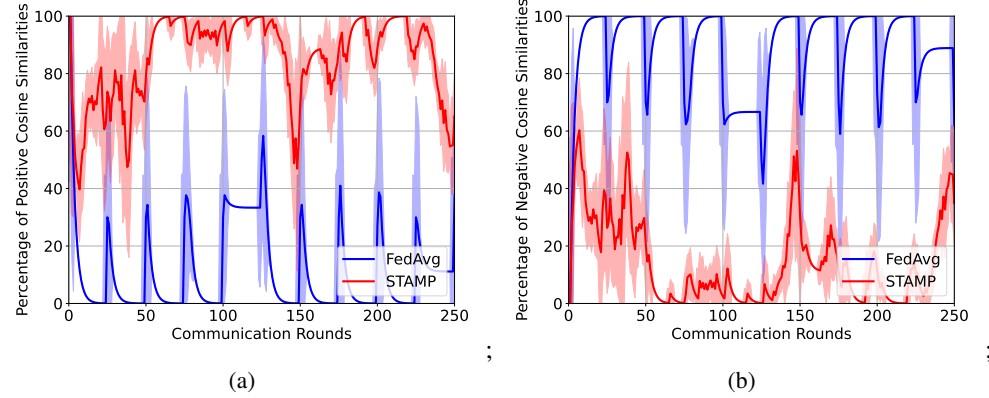

(a)                                                          (b)

Figure 22: Comparison between STAMP and FedAvg in terms of gradient alignment. STAMP significantly reduces gradient conflicts in a 10-client FL system.

### F.5.2 CASE OF 100 CLIENTS

To further validate the trend at a larger scale, we repeat the experiment using an FL setup with 100 clients. A client is again selected at random (denoted as client 1), and we compute the cosine similarities between its gradient and those of the remaining 99 clients during training. Figure 23 illustrates the gradient cosine similarities between client 1 and 9 representative clients chosen from the remaining pool. It is evident that STAMP consistently improves gradient alignment between client 1 and the selected peers, as indicated by higher cosine similarity values across training rounds.

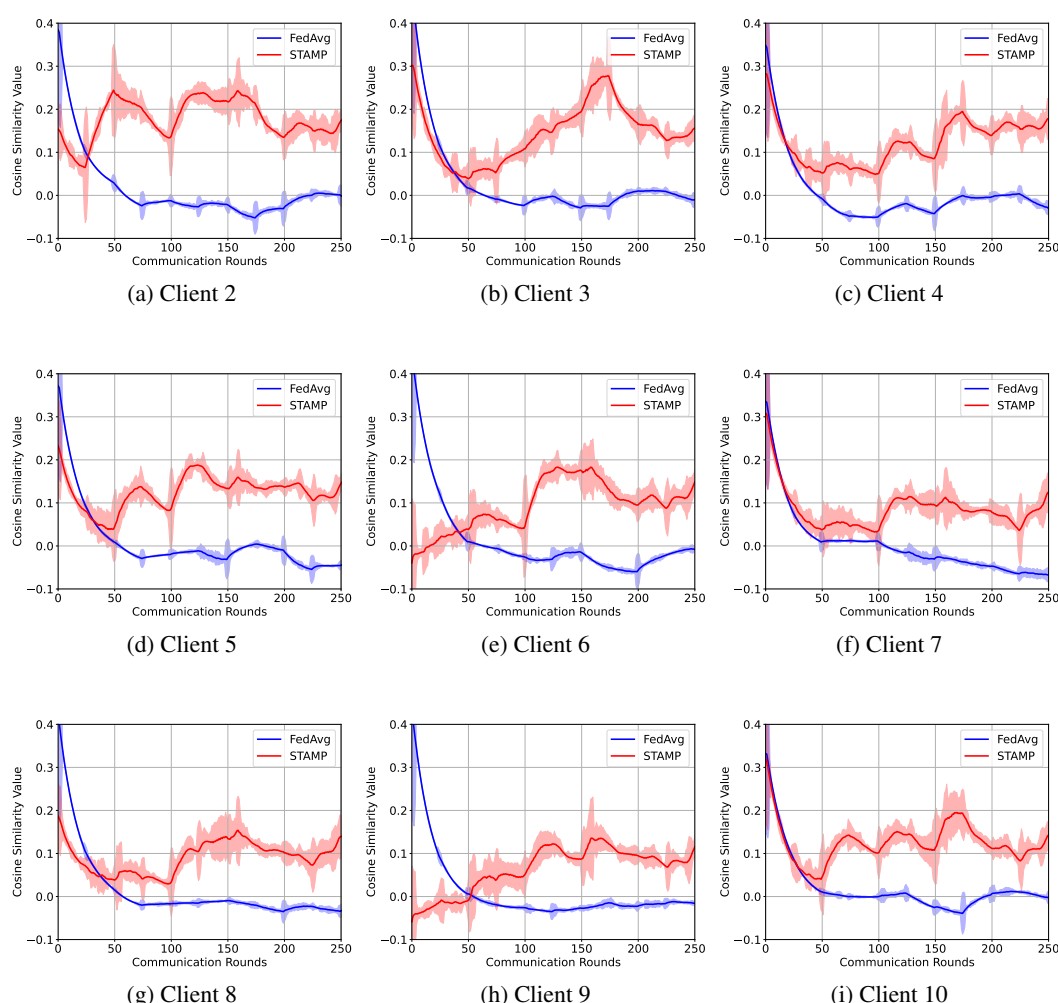

Figure 23: Cosine similarity between the gradient of client 1 and those of 9 selected clients in a 100-client FL system. STAMP improves alignment by increasing the cosine similarities across training rounds.

To summarize the overall trend across all clients, we count the number of positive and negative cosine similarities between client 1 and the other 99 clients at each training round. As shown in Figure 24a and Figure 24b, under FedAvg, client 1's gradient conflicts with more than 60% of the other clients for most of the training process. In contrast, STAMP significantly reduces the prevalence of gradient conflicts, lowering the proportion of negative similarities to below 10% in most rounds.

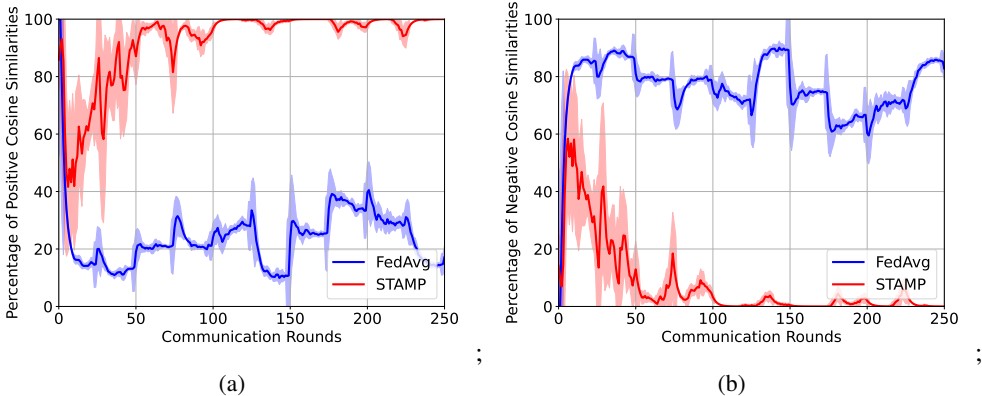

(a)                                                          ;                        (b)                                                          ;

Figure 24: Comparison between STAMP and FedAvg in aligning gradients in a 100-client FL system. STAMP significantly reduces gradient conflicts and increases agreement among client updates.

## F.6 EFFECTIVENESS OF PROTOTYPICAL CORESET

### F.6.1 EFFECTIVESS OF PROTONET

To evaluate the impact of prototypical coreset selection in STAMP with and without ProtoNet, we conduct an ablation study, with the results presented in Table 6. To further investigate why ProtoNet improves performance, we analyze the gradient alignment and its variance for STAMP and STAMP without ProtoNet, as shown in Figure 25. Two key observations emerge from Figure 25: first, STAMP without ProtoNet exhibits higher gradient variance; second, its gradient angles are lower compared to the full version. This can be attributed to the fact that higher gradient variance leads to less accurate gradient alignment.

Table 6: Ablation studies of the efectiveness of ProtoNet.

| Method | S-CIFAR-100 Acc ↑ | S-ImageNet1K Acc ↑ |
|---|---|---|
| ProtoNet | 52.8±0.9 | 41.5±2.8 |
| w/o ProtoNet | 47.6±0.8 | 36.3±1.3 |

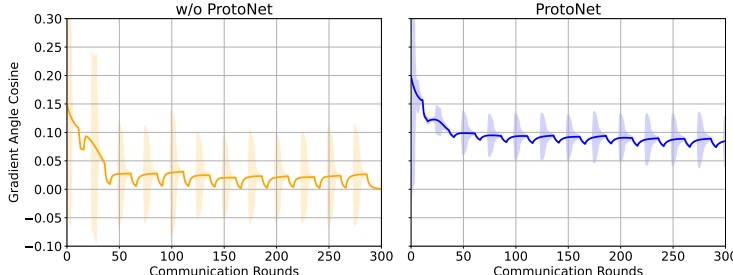

Figure 25: Effectiveness of ProtoNet on gradient angle. This is done on dataset S-ImageNet1K

### F.6.2 T-SNE VISUALIZATIONS

Figure 26 illustrates the effectiveness of prototype learning from a prototypical coreset. This figure highlights two key observations: (1) the inability of vanilla FL to effectively learn prototypes from hidden representations, and (2) the improved prototype learning capability achieved by STAMP. In the case of FedAvg, the model fails to acquire sufficiently representative features due to the limitations imposed by the single-pass data stream.

In contrast, STAMP demonstrates strong class discrimination as it progresses through tasks, which enhances its ability to learn prototypes from a compact coreset. This improvement stems from the coreset selection process, which is guided by class-specific criteria. As a result, it reduces inter-class confusion that could otherwise lead to inaccurate or misleading prototype representations.

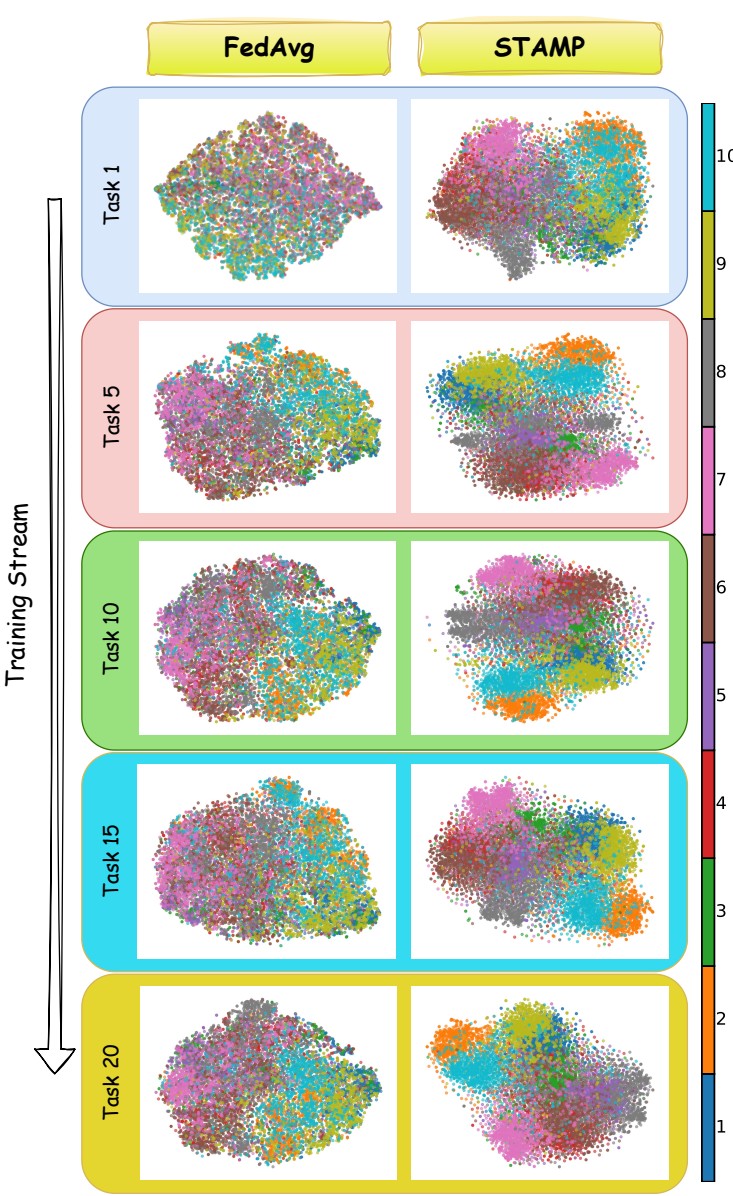

Figure 26: t-SNE visualizations of features learned by FedAvg and STAMP on the CIFAR-10 test set reveal notable differences. FedAvg exhibits significant class confusion when learning new classes, likely due to shortcut learning. In contrast, STAMP, leveraging a prototypical coreset, effectively mitigates forgetting and maintains clearer class separation.

## F.7 ADDITIONAL EVALUATIONS ON STAMP WITH VARYING CLIENT NUMBER

Table 7: Evaluation of STAMP and FedAvg on S-CIFAR100 and S-ImageNet1K datasets with 2 classes per client task. The experiments are conducted under different numbers of clients to assess scalability. Each result is averaged over 5 runs with standard deviation.

| | | **S-CIFAR100** $(C = 2)$ | | | |
|---|---|---|---|---|---|
| **Method** | **Metric** | **10** | **20** | **50** | **100** |
| FedAvg | Acc. | 31.7 $(\pm 1.7)$ | 26.8 $(\pm 1.9)$ | 16.2 $(\pm 2.5)$ | 8.8 $(\pm 2.9)$ |
| | AF | 22.1 $(\pm 1.3)$ | 20.3 $(\pm 0.9)$ | 13.7 $(\pm 1.7)$ | 6.8 $(\pm 1.1)$ |
| STAMP | Acc. | 52.8 $(\pm 0.9)$ | 48.3 $(\pm 0.6)$ | 41.7 $(\pm 1.1)$ | 31.4 $(\pm 0.9)$ |
| | AF | 24.3 $(\pm 0.8)$ | 23.3 $(\pm 0.4)$ | 20.5 $(\pm 0.8)$ | 18.4 $(\pm 0.9)$ |
| | | **ImageNet1K** $(C = 2)$ | | | |
| FedAvg | Acc. | 24.3 $(\pm 5.1)$ | 17.6 $(\pm 4.3)$ | 10.7 $(\pm 6.7)$ | 4.8 $(\pm 3.7)$ |
| | AF | 19.6 $(\pm 0.1)$ | 15.3 $(\pm 0.3)$ | 8.9 $(\pm 0.4)$ | 4.1 $(\pm 0.2)$ |
| STAMP | Acc. | 41.5 $(\pm 2.8)$ | 38.8 $(\pm 1.9)$ | 33.1 $(\pm 1.3)$ | 24.4 $(\pm 1.1)$ |
| | AF | 24.2 $(\pm 0.8)$ | 22.8 $(\pm 0.6)$ | 18.9 $(\pm 0.3)$ | 15.1 $(\pm 0.4)$ |

Table 7 presents an additional evaluations of the STAMP framework under varying numbers of clients (10, 20, 50, 100) on two benchmark datasets: S-CIFAR100 and S-ImageNet1K, with 2 classes per task. Across both datasets, as the number of clients increases, performance degrades for both methods due to increased heterogeneity and gradient conflicts. However, STAMP consistently outperforms FedAvg in all configurations, demonstrating stronger robustness and scalability. Notably, STAMP achieves higher accuracy with lower forgetting, especially in more challenging settings with a large number of clients.

## F.8 ADDITIONAL EVALUATIONS ON STAMP UNDER DIFFERENT PARTIAL PARTICIPATION RATES

Table 8: Performance of FedAvg and STAMP with 10 clients under different partial participation rates. STAMP is designed to remain robust as participation decreases.

| Method | Metric | 0.1 | 0.2 | 0.5 | 1.0 |
|--------|--------|-----|-----|-----|-----|
| **S-CIFAR100** ($C = 2$) | | | | | |
| FedAvg | Acc. | 22.5 ($\pm$ 2.8) | 26.1 ($\pm$ 2.3) | 29.7 ($\pm$ 1.8) | 31.7 ($\pm$ 1.7) |
| | AF | 16.4 ($\pm$ 1.9) | 18.2 ($\pm$ 1.4) | 20.5 ($\pm$ 1.1) | 22.1 ($\pm$ 1.3) |
| STAMP | Acc. | 45.3 ($\pm$ 1.6) | 48.0 ($\pm$ 1.2) | 51.0 ($\pm$ 1.0) | 52.8 ($\pm$ 0.9) |
| | AF | 20.8 ($\pm$ 1.0) | 22.1 ($\pm$ 0.8) | 23.5 ($\pm$ 0.6) | 24.3 ($\pm$ 0.8) |
| **ImageNet1K** ($C = 2$) | | | | | |
| FedAvg | Acc. | 13.8 ($\pm$ 6.5) | 16.3 ($\pm$ 5.4) | 20.9 ($\pm$ 4.7) | 24.3 ($\pm$ 5.1) |
| | AF | 10.8 ($\pm$ 1.1) | 13.7 ($\pm$ 0.9) | 17.3 ($\pm$ 0.6) | 19.6 ($\pm$ 0.1) |
| STAMP | Acc. | 33.7 ($\pm$ 3.9) | 36.9 ($\pm$ 3.2) | 39.8 ($\pm$ 2.6) | 41.5 ($\pm$ 2.8) |
| | AF | 19.1 ($\pm$ 1.2) | 21.0 ($\pm$ 0.9) | 23.1 ($\pm$ 0.6) | 24.2 ($\pm$ 0.8) |

Table 8 shows FedAvg is heavily impacted by low partial client participation. In contrast, STAMP remains substantially more robust thanks to temporal gradient alignment and prototypical coreset selection mechanism. This robustness becomes more pronounced as more clients participate, where STAMP consistently outperforms FedAvg by a large margin across both S-CIFAR100 and S-ImageNet1K. This is thanks to the proposed spatio gradient alignment.

## G   TIME COMPLEXITY OF PROTOTYPICAL CORESET SELECTION

**Theorem 3 (Time Complexity of Prototypical Coreset Selection)** *Under standard assumptions, Algorithm 2 has time complexity $\mathcal{O}(E \cdot C_\phi \cdot (|\mathcal{M}| + |\mathcal{N}^t|))$, where $L$ is the number of classes, $E$ is the number of epochs, $|\mathcal{M}^l|$ is the memory size per class, $|\mathcal{N}_l^t|$ is the number of new samples per class at task $t$, and $C_\phi$ is the computational cost of the encoder forward pass.*

*Proof.* Let $m = |\mathcal{M}^l|$, $n = |\mathcal{N}_l^t|$, we analyze the time complexity by examining each step of Algorithm 2 for a single class $l$, then aggregate over all $L$ classes.

**Step 1 (Prototype Computation):** Computing $g(x_i; \phi)$ for each $x_i \in \mathcal{N}_l^t$ requires $n$ encoder forward passes. The summation and normalization operations over $d$-dimensional vectors require $\mathcal{O}(nd)$ arithmetic operations. For instance,

$$\mathcal{T}_1 = \mathcal{O}(n \cdot C_\phi + nd). \tag{45}$$

**Step 2 (Initialization):** Initializing $|A| = n$ coefficients requires:

$$\mathcal{T}_2 = \mathcal{O}(n). \tag{46}$$

**Step 3 (Optimization Loop):** For each epoch $e \in \{1, \ldots, E\}$:

- Computing embeddings for samples in $\mathcal{M}^l$ requires $m$ encoder forward passes: $\mathcal{O}(m \cdot C_\phi)$
- Computing embeddings for samples in $\mathcal{N}_l^t$ requires $n$ encoder forward passes: $\mathcal{O}(n \cdot C_\phi)$
- Computing weighted sums $\sum_{i \in \mathcal{M}^l} g(x_i; \phi)$ and $\sum_{i \in \mathcal{N}_l^t} a_i g(x_i; \phi)$ requires $\mathcal{O}((m + n)d)$ operations
- Computing the squared norm requires $\mathcal{O}(d)$ operations
- Computing gradient $\nabla_A \mathcal{L}_{\texttt{proto}}$ and updating $A$ requires $\mathcal{O}(n)$ operations

Over $E$ epochs, we have the following time complexity:

$$\mathcal{T}_3 = \mathcal{O}(E \cdot [(m + n) \cdot C_\phi + (m + n)d + n]) \tag{47}$$

The Top-$k$ selection can be implemented using quickselect in expected $\mathcal{O}(n)$ time or heap-based selection in $\mathcal{O}(n \log m)$ time:

$$\mathcal{T}_{3,\text{select}} = \mathcal{O}(n \log m) \tag{48}$$

**Step 4 (Style Mixing):** In the worst case, applying MixStyle to $m$ samples requires:

$$\mathcal{T}_4 = \mathcal{O}(m \cdot C_{\text{mix}}) \tag{49}$$

**Step 5 (Memory Update):** Updating the memory requires:

$$\mathcal{T}_5 = \mathcal{O}(m) \tag{50}$$

Combining all steps, we have the following computation complexity as follows:

$$\begin{aligned}
\mathcal{T}_{\text{class}} &= \mathcal{T}_1 + \mathcal{T}_2 + \mathcal{T}_3 + \mathcal{T}_{3,\text{select}} + \mathcal{T}_4 + \mathcal{T}_5 \\
&= \mathcal{O}\big(n \cdot C_\phi + nd + n + E \cdot [(m + n) \cdot C_\phi + (m + n)d + n] \\
&\quad + n \log m + m \cdot C_{\text{mix}} + m\big)
\end{aligned} \tag{51}$$

To simplify the computation complexity, we follow the following assumptions typical in continual learning settings:

**Assumption 4 (Encoder Dominance)** *The encoder forward pass dominates other operations: $C_\phi \gg d, C_{mix}$.*

**Assumption 5 (Multiple Epochs)** *The number of optimization epochs satisfies $E \geq 1$, typically $E \gg 1$ in practice.*

Under Assumptions 4 and 5, the dominant term in $\mathcal{T}_{\text{class}}$ is $E \cdot (m + n) \cdot C_\phi$ because we have the three following statements:

$$E \cdot (m + n) \cdot C_\phi \gg n \cdot C_\phi \quad \text{(for } E \geq 1 \text{ and } m > 0) \tag{52}$$

$$E \cdot (m + n) \cdot C_\phi \gg E \cdot (m + n) \cdot d \quad \text{(by Assumption 4)} \tag{53}$$

$$E \cdot (m + n) \cdot C_\phi \gg n \log m, m \cdot C_{\text{mix}} \quad \text{(by Assumption 4)} \tag{54}$$

As a consequence, we have the following simplified complexity for each class as follows:

$$\mathcal{T}_{\text{class}} = \mathcal{O}(E \cdot (m + n) \cdot C_\phi) \tag{55}$$

Since the algorithm iterates over $L$ classes independently, we have the following total computational complexity as follows:

$$\mathcal{T}_{\text{total}} = L \cdot \mathcal{T}_{\text{class}} = \mathcal{O}(L \cdot E \cdot (m + n) \cdot C_\phi)$$
$$= \mathcal{O}(L \cdot E \cdot (|\mathcal{M}^l| + |\mathcal{N}_l^t|) \cdot C_\phi) = \mathcal{O}(E \cdot (|\mathcal{M}| + |\mathcal{N}^t|) \cdot C_\phi) \tag{56}$$

This completes the proof.

## H  PRIVACY OF STAMP

FL (McMahan et al., 2017), and FCL in particular, are vulnerable to various attacks such as data poisoning, model poisoning (Wan et al., 2024), backdoor attacks (Nguyen et al., 2023), and gradient inversion attacks (Petrov et al., 2024; Balunovic et al., 2022; Dimitrov et al., 2022). Our proposed method does not introduce any additional privacy risks beyond those inherent to the standard FedAvg algorithm. Consequently, it is compatible with existing defense mechanisms developed for FedAvg, including secure aggregation (Mai et al., 2024; So et al., 2023) and noise injection prior to aggregation (Hu et al., 2024).

Unlike several prior FCL approaches (Zhang et al., 2023b; Qi et al., 2023) that require clients to share either locally trained generative models or perturbed private data, STAMP relies solely on gradient alignment. It utilizes the global model weights and the uploaded local model updates, information already exchanged among clients in the standard FedAvg setting, thus avoiding the need for additional private data sharing, especially over open communication environments (e.g., 5G/6G wireless networks).

## I  LIMITATIONS AND FUTURE WORKS

A primary limitation of our method lies in the sensitivity of gradient alignment to the stability of task-wise and client-wise gradient trajectory approximation. Moreover, existing gradient alignment approaches typically learn a single parameter set that adjusts the magnitude of task-specific gradients through a convex combination. Such approaches do not influence the direction of the gradients. Therefore, enhancing the stability of gradient trajectory approximation and improving gradient alignment performance, particularly by extending the learnable parameter set to operate at the layer-wise or element-wise level, emerge as a promising direction for future research.

