# OpenReview forum: "Exploiting Client Heterogeneity for Forgetting Mitigation in Federated Continual Learning: A Spatio-Temporal Gradient Alignment Approach"
_ICLR.cc/2026/Conference — Submitted to ICLR 2026_

### Official Review · Reviewer_frwb · 2025-10-27

**Soundness:** 3
**Presentation:** 3
**Contribution:** 3
**Rating:** 6
**Confidence:** 5

**Summary:**

This paper proposes STAMP for FCL. It introduces temporal gradient alignment on clients and spatial gradient alignment on the server to handle task and client heterogeneity. At the same time, a prototypical coreset mechanism provides efficient replay without heavy memory usage. The method aims to align gradients across both temporal and spatial dimensions, improving generalization and reducing forgetting. Extensive experiments on diverse datasets demonstrate that STAMP outperforms existing FCL methods in accuracy, stability, and efficiency.

**Strengths:**

- Treating spatio-temporal gradient alignment jointly is a creative and well-motivated way to handle heterogeneity in FCL.
- The prototypical coreset approach is lightweight and avoids the heavy cost of generative replay, which is appealing for real-world edge deployments.
- The paper conducts comprehensive evaluations across multiple datasets and metrics, showing clear and consistent improvements.
- The inclusion of a theoretical generalization bound adds some rigor and helps contextualize the method’s stability and plasticity claims.

**Weaknesses:**

- The theoretical analysis remains largely qualitative and does not clearly demonstrate why STAMP achieves better gradient alignment in practice.
- While the proposed approach is efficient, the algorithmic pipeline (temporal + spatial alignment + coreset) can be conceptually heavy and may limit reproducibility.
- The improvements, although consistent, are relatively modest on some benchmarks, and the scalability on extremely large client pools is not deeply analyzed.

**Questions:**

- How sensitive is STAMP to the size of the coreset, and what is the trade-off between memory footprint and accuracy?
- Can STAMP handle asynchronous or partially participating clients in more realistic federated settings?
- How does STAMP perform when the degree of heterogeneity is extreme, e.g., when client label spaces are disjoint?
- Some related baselines like FedSSI and references should be added and compared [1-2].

[1] FedSSI: Rehearsal-Free Continual Federated Learning with Synergistic Synaptic Intelligence. The Thirteenth International Conference on Learning Representations, ICML 2025

[2] Unleashing the Power of Continual Learning on Non-Centralized Devices: A Survey. IEEE Communications Surveys & Tutorials, 2025

---

> ### Author Response · Authors · 2025-11-25
>
> We thank the Reviewer for a careful consideration into our paper, and consider highly of our motivation and technique used in our paper. We have thoroughly considered all comments and have revised the manuscript accordingly. The updated version will be released once we have fully addressed and incorporated the feedback from all Reviewers.
>
> 1. **Why STAMP achieve better gradient alignment:** To understand why STAMP achieves better gradient alignment in practice, we have provided the extended both theoretical proof on the gradient alignment and experimental evaluations on the catastrophic forgetting.
> The theoretical proof on gradient alignment to show why it can reduce the gradient conflicts is demonstrated in Appendix B.
> The evaluations show that the gradient angles aligned with the forgetting curves. Specifically, as the angles are higher, the forgetting curves less steep angle.
> To evaluate the scalability of STAMP, we conduct experiments using variation client pools (10, 20, 50, 100 clients) and demonstrate in Appendix F.7.
> ### CIFAR100 (C = 2)
>
> | Method | Metric | 10 | 20 | 50 | 100 |
> |--------|---------|---------|---------|---------|---------|
> | FedAvg | Acc. | 31.7 (± 1.7) | 26.8 (± 1.9) | 16.2 (± 2.5) | 8.8 (± 2.9) |
> |        | AF    | 22.1 (± 1.3) | 20.3 (± 0.9) | 13.7 (± 1.7) | 6.8 (± 1.1) |
> | STAMP  | Acc. | 52.8 (± 0.9) | 48.3 (± 0.9) | 41.7 (± 1.1) | 31.4 (± 0.9) |
> |        | AF    | 24.3 (± 0.8) | 23.3 (± 0.4) | 20.5 (± 0.8) | 18.4 (± 0.9) |
>
>
> ### ImageNet1K (C = 2)
>
> | Method | Metric | 10 | 20 | 50 | 100 |
> |---------|---------|---------|---------|---------|---------|
> | FedAvg  | Acc. | 24.3 (± 5.1) | 17.6 (± 4.3) | 10.7 (± 6.7) | 4.8 (± 3.7) |
> |         | AF    | 19.6 (± 0.1) | 15.3 (± 0.3) | 10.0 (± 4.9) | 4.4 (± 0.7) |
> | STAMP   | Acc. | 41.5 (± 2.8) | 38.8 (± 1.9) | 33.1 (± 3.4) | 24.4 (± 1.1) |
> |         | AF    | 24.2 (± 0.8) | 22.8 (± 0.6) | 18.9 (± 0.3) | 15.1 (± 0.4) |
>
> 2. **How sensitive is STAMP to the size of coreset:** We have already analyzed the sensitivity of STAMP to the coreset size in the original manuscript and presented in Section 5.2. Specifically, STAMP demonstrates that it can achieve competitive performance with only 20 samples per class. This number of samples per class is generally considered sufficient in rehearsal-based continual learning, and in federated continual learning in particular.
> 3. **Realistic federated settings:** To analyze the performance of STAMP under partially participating clients, we have conducted additional experiments. The additional results can be found in Appendix F.8.
> ### CIFAR100 (C = 2)
>
> | Method | Metric | 0.1 | 0.2 | 0.5 | 1.0 |
> |--------|---------|---------|---------|---------|---------|
> | FedAvg | Acc. | 22.5 (± 2.8) | 26.1 (± 2.3) | 29.7 (± 1.8) | 31.7 (± 1.7) |
> |        | AF    | 16.4 (± 1.9) | 18.2 (± 1.4) | 20.5 (± 1.1) | 22.1 (± 1.3) |
> | STAMP  | Acc. | 45.3 (± 1.6) | 48.0 (± 1.2) | 51.0 (± 1.0) | 52.8 (± 0.9) |
> |        | AF    | 20.8 (± 1.0) | 22.1 (± 0.8) | 23.5 (± 0.6) | 24.3 (± 0.8) |
>
>
> ### ImageNet1K (C = 2)
>
> | Method | Metric | 0.1 | 0.2 | 0.5 | 1.0 |
> |---------|---------|---------|---------|---------|---------|
> | FedAvg  | Acc. | 13.8 (± 6.5) | 16.3 (± 5.4) | 20.9 (± 4.7) | 24.3 (± 5.1) |
> |         | AF    | 10.8 (± 1.1) | 13.7 (± 0.9) | 17.3 (± 0.6) | 19.6 (± 0.1) |
> | STAMP   | Acc. | 33.7 (± 3.9) | 36.9 (± 3.2) | 39.8 (± 2.6) | 41.5 (± 2.8) |
> |         | AF    | 19.1 (± 1.2) | 21.0 (± 0.9) | 23.1 (± 0.6) | 24.2 (± 0.8) |
>
> 4. **Degree of heterogeneity:** In the original version of the manuscript, we already evaluated STAMP together with other baselines under varying degrees of heterogeneity. In particular, we included experiments under highly heterogeneous settings, corresponding to Dirichlet coefficients $\alpha = 0.1$ and $\alpha = 1.0$. These conditions reflect extremely heterogeneous task and data distributions across clients.
>
> 5. **Literature reviews:** In our original manuscript, we have already included FedSSI as one of the baseline methods, and its results are reported in Figure 1. We also discuss the FedSSI paper [1] in the Related Works section to provide a more comprehensive review of the existing literature. In addition, we have incorporated ReFed+ [2] into our experimental evaluation, including both its implementation and results. The implementations of FedSSI and ReFed+ will be publicly released alongside the main repository upon the paper’s publication. We also discussed about [3] in our paper to improve the comprehensiveness of the literature reviews.
>  - [1] FedSSI: Rehearsal-Free Continual Federated Learning with Synergistic Synaptic Intelligence. The Thirteenth International Conference on Learning Representations, ICML 2025.
>  - [2] Re-Fed+: A Better Replay Strategy for Federated Incremental Learning, IEEE TPAMI, 2025.
>  - [3] Unleashing the Power of Continual Learning on Non-Centralized Devices: A Survey. IEEE Communications Surveys & Tutorials, 2025.

---

> ### Author Response · Authors · 2025-11-28
> **Following-up comments**
>
> Dear Reviewer frwb,
>
> We would like to gently follow up on our earlier rebuttal. We understand the considerable workload during the review process and sincerely appreciate the time you devote to evaluating submissions.
>
> We would also be grateful if you could take a moment to look at the additional elements included in our updated response, namely, the new experimental results and our expanded theoretical analysis in the gradient alignment. Any feedback or comments you might have on these points would be greatly appreciated.
>
> Thank you very much for your time and consideration.
>
> Best regards,
>
> Authors

---

### Official Review · Reviewer_Ya9R · 2025-10-28

**Soundness:** 3
**Presentation:** 2
**Contribution:** 3
**Rating:** 6
**Confidence:** 4

**Summary:**

The paper proposes the use of gradient alignment technique across tasks and clients computed from a prototype coreset of samples (proposed in this paper) at each client to improve generalization in federated continual learning setting. The paper also provides bounds on the generalization gap between seen and unseen tasks.

**Strengths:**

- The paper provides theoretical justification for gradient alignment techniques and specifically for the coreset selection technique being used
- Generalization bound is derived and proposed technique is shown to have reduced the generalization gap.
- Performance is shown to be on par with popular prior work such as FedWeIT, while reducing the disk usage significantly.
- Ablation studies are quite extensive, the results shown are convincing to show that the method work reliably and provides the claimed gains. Datasets however are somewhat still more traditional datasets used in federated learning and does not contain natural continual learning datasets.

**Weaknesses:**

- The method relies on some assumptions such as requiring replay buffer at clients.
- Gradient alignment might slow the convergence, especially when updating the global model.
- The paper should at least in the appendix comment on how the coreset selection (combinatorial problem) is solved and what’s the additional worst-case complexity (O(n^2) or O(n log n), etc).
- The claims are a bit overstated as the generalization over tasks is not really the goal of the coreset selection. For example, assume a large volume of samples from a certain task/client that dominates a few training rounds. In those cases, the coreset will start to get dominated by samples from those tasks. As a result, the gradient alignment favors those tasks containing the most samples. Instead of task level, I suppose the claim is that the proposed technique will help in reducing generalization gap across the diverse data.

**Questions:**

- How is the coreset selection done? Even if this is a well-known and solved problem, it should clearly be stated including the computational complexity of it.
- Is the paper's goal really the generalization over all data or tasks? Check comments above, but the question is what happens when a single task has 1000x more samples compared to another task?

---

> ### Author Response · Authors · 2025-11-25
>
> We thank the Reviewer for recognizing the contributions of our work and for the constructive feedback. Below, we provide detailed responses to the comments, focusing on the aspects of data, convergence, learning complexity, and model generalization, as well as addressing the specific concerns raised. We will update the revised manuscript once all Reviewers' comments have been fully addressed.
> 1. **Datasets does not contain continual characteristics:** In our work, we employ S-CIFAR100 and S-ImageNet1K, both of which are widely used datasets that exhibit natural properties suitable for continual learning. In response to the reviewer’s comment, we will revise the manuscript to clarify the dataset configurations and to ensure that their roles in our experimental design are explicitly articulated. We have also updated the dataset descriptions to S-CIFAR100 [1] and S-ImageNet1K [2], which correspond to the two settings discussed in the original version of the manuscript.
>  - [1]: Class-Incremental Continual Learning into the eXtended DER-verse, IEEE TPAMI 2025.
>  - [2]: Loss of plasticity in deep continual learning, Nature 2024.
> 2. **Slow convergence of the gradient alignment:** Based on our limited understanding, we believe the Reviewer is referring to convergence time in terms of wall-clock seconds, which may be affected by the computational cost of gradient alignment. We infer this interpretation because gradient alignment has already been shown to be robust with respect to convergence per communication round in both standard FL and FCL settings. We agree that in several prior works (GEM or A-GEM), updating gradient alignment can be computationally expensive, where the optimization is applied to learnable parameters of gradients $d$. For instance,
> $$
>   \min_{\widetilde{g}} \frac{1}{2} \Vert g - \widetilde{g} \Vert^2_2 \quad \textrm{s.t.} \quad \langle \widetilde{g}, g_k \rangle \geq 0, \quad \forall k<t, \quad \widetilde{g}, g \in \mathbb{R}^d
> $$
> We address this concern by applying alignment only to a set of learnable weights whose dimensionality equals the number of users $T$ ($T \ll d$).
> $$
> \Gamma^* = \arg\min_{\Gamma} \Gamma\mathbf{g}^{(r)}\cdot \widetilde{g}^{(r)} + \kappa\Vert \widetilde{g}^{(r)}\Vert\Vert g^{(r)}_{\Gamma}\Vert,
> \quad \Gamma \in \mathbb{R}^T,
> \quad \widetilde{g} = \frac{1}{T}\sum g_t
> $$
> As a result, our method incurs substantially lower computational overhead than approaches such as GEM or A-GEM. To substantiate this claim, we have included a complexity analysis of our gradient alignment mechanism and revised the manuscript to emphasize the computational efficiency of the proposed approach.
> 3. **Coreset selection complexity:** According to the Reviewer’s suggestion, we have provided the complexity analysis in the Appendix H. Specifically, our proposed coreset selection method has time complexity $\mathcal{O}(E \cdot  C_\phi \cdot (\vert\mathcal{M}\vert+\vert\mathcal{N}^t\vert))$, where $L$ is the number of classes, $E$ is the number of epochs, $\vert\mathcal{M}^l\vert$ is the memory size, $\vert\mathcal{N}^t\vert$ is the number of new samples at task $t$, and $C_\phi$ is the computational cost of the encoder forward pass.
>
> 4. **Overstated claim of generalization with Coreset Selection:** We apologize for any misunderstanding regarding the contributions of our work. As stated in the paper, the prototypical coreset is designed to approximate gradients without requiring stored historical gradients. Storing past gradients does not reflect the gradient landscape under the current model parameters, which can reduce generalization when applying gradient matching. Our approach addresses this by using coresets that dynamically capture the model’s present behavior, thereby enabling more reliable gradient estimation and improved alignment.

---

> ### Author Response · Authors · 2025-11-28
> **Following-up comment**
>
> Dear Reviewer Ya9R,
>
> We would like to kindly follow up on our earlier rebuttal. We fully understand the significant workload during the review period and sincerely appreciate the time you invest in assessing the submissions.
>
> If you have the opportunity, we would be grateful if you could also consider the additional components included in our updated response, i.e., the new experimental results and the extended theoretical analysis of the gradient alignment and coreset selection. We would greatly appreciate any feedback you may have on these aspects.
>
> Thank you again for your time and consideration.
>
> Best regards,
>
> Authors

---

### Official Review · Reviewer_CrLV · 2025-10-31

**Soundness:** 3
**Presentation:** 2
**Contribution:** 3
**Rating:** 4
**Confidence:** 4

**Summary:**

This paper proposes STAMP, a federated continual learning (FCL) approach incorporating spatio-temporal gradient alignment across clients and tasks, along with a prototypical coreset to mitigate catastrophic forgetting with reduced memory usage. The idea of aligning gradients across the temporal (intra-client) and spatial (inter-client) dimensions is interesting, and the paper provides both theoretical motivation and empirical evaluations showing improvement over existing baselines on several vision datasets.

**Strengths:**

1.Novel idea of performing both temporal and spatio gradient alignment in FCL.
2.Prototypical coreset is a good alternative to generative replay and full memory buffers.

**Weaknesses:**

- Handling catastrophic forgetting: Although this work is rehearsal based, the work does not explicitly handle catastrophic forgetting (most of the rehearsal methods do, such as AGEM/GEM). However, the paper repeatedly claims that STAMP reduces catastrophic forgetting, but this is not explicitly demonstrated experimentally or theoretically. The provided plots focus only on temporal and spatio gradient alignment metrics. Good alignment does not necessarily guarantee reduced forgetting. More direct evidence such as forgetting curves and class/task-level retention is needed.
- The claim of Fig 1 is misleading. The difference indeed decreases, but at the cost of global accuracy in STAMP. How is this a good case? This also contradicts the claim of improved intra-client retention. The paper needs to explain this discrepancy.
- It is unclear how spatio and temporal gradient alignment preserve the gradient direction, especially in coparison with the memory data. The paper states that alignment prevents negative transfer, but lacks intuition or analysis on how this specifically preserves directionality of task gradients over time.
- It is not clear how the theoretical results show the impact of using coresets - a typical generalization result must encompass the effect of coresets. For instance, if one performed random sampling of points instead of using a coreset, how would the effect be reflected in Thm 2.
- Storage cost concern for storing gradients. Calculating gradient alignment implicitly requires storing gradients from previous tasks and clients. This may become costly for large models.
- Ablation studies are insufficient. The method introduces multiple components (temporal GA, spatio GA, prototypical coreset, ProtoNet, MixStyle), yet ablations are limited. Add more like varying number of tasks, effect of different epochs per task, impact of removing the prototypical network, varying coreset sizes.
- \gamma used in the gradient alignment formulation is not defined in the main text.
- Possible error in Section 2.1. The description states that r is the current round of task t, but it should logically refer to the current round of task t+1.
- Figure 3 does not support the claim that heterogeneity helps generalization. The results do not show a clear benefit as heterogeneity increases. This contradicts the core hypothesis. The chosen values of \alpha are quite high, and does not indicate highly heterogeneous cases.
Missing plots in Figures 5 and 6. No results for CIFAR100 with 2 classes/task for temporal gradient alignment in Fig. 5. No results for CIFAR100 with 20 classes/task for spatio gradient alignment Fig. 6. Some recent baselines with theoretical guarantees (CFLAG, AISTATS 2025) is not cited.

**Questions:**

Please clarify the points raised in weaknesses.

---

> ### Author Response · Authors · 2025-11-26
>
> 1. **Handling catastrophic forgetting**: We thank the Reviewer for the constructive comments. Following the Reviewer’s suggestion, we conducted additional experiments and provided a discussion on the relationship between gradient alignment in Section 5.2; detailed analyses are presented in the Appendix F.3. The results are as follows:
> ### S-CIFAR100 (C = 2)
> | Task | FedAvg | FedL2P | AF-FCL | FedCIL | TARGET | FedSSI | STAMP |
> |------|--------|--------|--------|--------|--------|--------|--------|
> | 2    | 46.11  | 50.32  | 51.41  | 49.02  | 50.77  | 54.18  | 52.36  |
> | 4    | 41.22  | 44.67  | 49.58  | 45.89  | 47.41  | 50.91  | 52.84  |
> | 6    | 37.88  | 45.03  | 44.19  | 40.72  | 46.33  | 48.01  | 50.72  |
> | 8    | 31.63  | 41.42  | 38.51  | 35.61  | 42.77  | 41.33  | 49.02  |
> | 10   | 26.91  | 36.44  | 36.42  | 34.03  | 39.33  | 40.71  | 48.61  |
> | 12   | 22.04  | 32.71  | 29.44  | 31.49  | 35.52  | 33.44  | 47.38  |
>
> ### S-ImageNet1K (C = 2)
> | Task | FedAvg | FedL2P | AF-FCL | TARGET | FedCIL | FedSSI | STAMP |
> |------|--------|--------|--------|--------|--------|--------|--------|
> | 2    | 41.22  | 44.61  | 45.51  | 46.41  | 41.91  | 48.54  | 45.31  |
> | 4    | 38.01  | 39.73  | 40.82  | 42.33  | 40.44  | 44.88  | 44.37  |
> | 6    | 31.12  | 36.77  | 34.33  | 37.18  | 38.03  | 38.74  | 43.02  |
> | 8    | 27.42  | 31.14  | 27.74  | 31.53  | 28.61  | 34.22  | 42.11  |
> | 10   | 23.33  | 26.41  | 24.71  | 27.91  | 28.04  | 30.72  | 41.44  |
> | 12   | 15.73  | 22.88  | 18.61  | 21.34  | 21.02  | 24.11  | 38.62  |
>
> From a theoretical perspective, our results demonstrate that STAMP mitigates catastrophic forgetting by minimizing the temporal task statistical distance on each client. This is aligned with the experimental evaluations, where higher gradient angles between tasks correspond to more gradual decline in the forgetting curves, indicating less catastrophic forgetting.
>
> 2. **Explanation on Figure 1**: We thank the Reviewer for the constructive comment.
> First, we would like to clarify that in STAMP, the global accuracy does not decrease as a consequence of reducing the generalization gap. The observed decrease occurs in the local accuracy, not the global one. Typically, local accuracy tends to be high because many existing works prioritize improving each client’s personalized performance. However, enhancing personalization often introduces a trade-off with generalization. This explains why, in our results, local accuracy may decrease when the generalization gap is reduced.
> Second, improving intra-client retention must be accompanied by strengthening global generalization in heterogeneous FCL. When a method focuses solely on retaining each client’s past knowledge, the model becomes increasingly tailored to that client’s non-IID distribution, leading to local overfitting. This over-specialization reduces the compatibility of local updates with the global model and limits the overall ability to incorporate and transfer useful information across clients. Therefore, effective FCL requires jointly preserving local knowledge and promoting generalizable representations that support cross-client learning. In response to the Reviewer’s comment, we have revised the manuscript and expanded the discussion to provide a clearer explanation of Figure 1.
> 3. **Justification on gradient alignment:** We thank the Reviewer for the constructive comment. In response, we have added a theoretical analysis that explains why gradient alignment mitigates negative transfer. In general, the motivation come from considering the worse generalization case among all seen tasks $t\in \mathcal{T}$ with current model $\theta^{(r)}$.
> $$
> \mathtt{GAP}(\theta, x)
> = \max\_{t \in \mathcal{T}} \big[ \frac{1}{\eta} \mathcal{L}(\theta^{(r)} - \eta x; \mathcal{D}^t\_u) - \mathcal{L}(\theta^{(r)}; \mathcal{D}^t\_u) \Big]
> = \max\_{t \in \mathcal{T}} \Big[-\nabla_{\theta}\mathcal{L}(\theta^{(r)};\mathcal{D}^t_u)^{\top} x + \frac{\eta}{2} x^{\top} \nabla^2_{\theta}\mathcal{L}(\theta^{(r)};\mathcal{D}^t_u)(\theta^{(r)}) x + \mathcal{O}(\eta) \Big]
> $$
> Therefore, it can be reduced to
> $$
> \mathtt{GAP}(\theta, x)
> \approx \max\_{t \in \mathcal{T}} \Big[-\nabla_{\theta}\mathcal{L}(\theta^{r};\mathcal{D}^t_u)^{\top} x \Big] \approx - \min\_{t\in\mathcal{T}}\langle \nabla_{\theta}\mathcal{L}(\theta^{r};\mathcal{D}^t_u), x \rangle \approx - \min\_{t\in\mathcal{T}}\langle g^{(t,r)}\_u, x \rangle
> $$
> Then, we consider it as a optimization problem
> $$
> \max\_{x} \min\_{\lambda, \gamma} ~ (\sum\_{u\in\mathcal{U}_\mathcal{S}} \gamma_u g^{(r)}\_u)^{\top} x - \frac{\lambda}{2}\Vert \bar{g}^{(r)} - x\Vert^2 + \frac{\lambda}{2} \phi,\quad \text{s.t. } \lambda\geq 0.
> $$
> By fixing each $(\lambda,\gamma)$ and deriving over $x$, we achieve the proposed gradient alignment. The detailed can be found in the Appendix B.

---

> ### Author Response · Authors · 2025-11-26
> **Official Comment by Authors (2)**
>
> 4. **Storage concerns by storing gradients:** We emphasize that our method does not store gradients for gradient alignment. Instead, we maintain a small coreset in memory to approximate the gradient trajectories. Consequently, the memory footprint remains modest, and the approach scales efficiently even for large models.
> 5. **Insufficient ablation studies:** We thank the Reviewer for the constructive comment. In our original work, we have already conducted an ablation study by removing the prototypical coreset selection module, as reported in Table 2. In this additional experiment, we evaluate the model without prototypical coresets by replacing them with reservoir sampling. The corresponding results are presented in the following table.
>
> | Dataset      | Metric | FedAvg        | (1)            | (2)            | (1) + (2)       | (1) + (3)       | (2) + (3)       | STAMP          |
> |--------------|--------|----------------|-----------------|-----------------|------------------|------------------|------------------|-----------------|
> | **S-CIFAR100** | Acc.   | 31.7 ± 1.7     | 38.1 ± 1.3      | 37.8 ± 0.6      | 44.7 ± 1.5       | 46.1 ± 0.7       | 44.9 ± 1.4       | 52.8 ± 0.9      |
> |              | AF     | 22.1 ± 1.3     | 23.8 ± 0.4      | 21.7 ± 0.9      | 21.5 ± 1.0       | 24.7 ± 1.4       | 21.8 ± 0.6       | 24.3 ± 0.8      |
> | **ImageNet1K** | Acc.   | 24.3 ± 5.1     | 30.5 ± 2.8      | 28.3 ± 2.6      | 34.1 ± 0.7       | 37.4 ± 1.1       | 36.5 ± 1.3       | 41.5 ± 2.8      |
> |              | AF     | 19.6 ± 0.1     | 26.1 ± 0.7      | 23.8 ± 0.6      | 24.3 ± 0.9       | 26.1 ± 1.8       | 23.3 ± 0.8       | 24.2 ± 0.8      |
>
> We acknowledge that the comment of the Reviewer according to the ablation study help us emphasize the paper's contribution and significance. As a consequence, we performed further experiments by adjusting the number of training epochs per task.
>
> | Epochs / Round | Last Accuracy (± Std) | Average Forgetting (± Std) |
> |-----------------|------------------------|------------------------------|
> | 15 | 35.98 (± 0.709) | 20.31 (± 0.754) |
> | 20 | 45.20 (± 0.955) | 23.03 (± 1.332) |
> | 25 | 52.80 (± 1.253) | 24.30 (± 0.640) |
> | 30 | 53.01 (± 0.754) | 29.90 (± 1.704) |
> | 35 | 51.25 (± 0.482) | 39.48 (± 1.274) |
>
> We also make ablation test according to the removing the prototypical network component to better understand the algorithmic contributions.
>
> | Method        | S-CIFAR-100 Acc | S-ImageNet1K Acc |
> |---------------|----------------|-----------------|
> | ProtoNet      | 52.8 ± 0.9     | 41.5 ± 2.8      |
> | w/o ProtoNet  | 47.6 ± 0.8     | 36.3 ± 1.3      |
>
> To further investigate why ProtoNet improves performance, we analyze the gradient alignment and its variance for STAMP and STAMP without ProtoNet and shown in Appendix F.6.1. The results
>
> In addition, we note that in the original manuscript, we conducted experimental evaluations of both temporal and spatio-temporal GA to assess their impact on learning performance. The results are shown in Figures 5 and 6, and Appendix F.5.2.
> We also performed experiments with varying coreset sizes to further analyze the method’s behavior in our original manuscript. The results are shown in figure 4.
>
> 6. **Gamma not defined:** We thank the Reviewer for the constructive comment. Specifically, $\gamma$ denotes the learnable weights used in the gradient aggregation process. We define $\Gamma = [\gamma_1, \ldots, \gamma_U]$, where $U$ is the number of users, and the global gradient is computed as $g_G = \sum_{u \in \mathcal{U}} \gamma_u g_u$. We have revised the manuscript accordingly and added a detailed discussion of the $\gamma$ variables in the figure.
>
> 7. **Description of task:** We thank the Reviewer for the constructive comment. Here, we provide clarification regarding our preliminaries. In our work, we denote the round index in each task as $r \in [0, R]$. When $r = R + 1$, it is reset to $r = 0$ and the next task begins. The logical flow of this procedure can be found in the code provided in the supplementary material.
>
> 8. **Discussion on Figure 3, missing plots on Figures 5 and 6:**
> We thank the Reviewer for the constructive comment. In response, we have added the addtional plots for Figures 5 and 6.
> According to figure 3, we would like to have discussion as follows. To investigate the impact of the choice of $\alpha$, we vary its value from high to low, thereby evaluating the performance of the proposed method under different degrees of data heterogeneity, ranging from low non-i.i.d. levels (closer to i.i.d.) to high non-i.i.d. levels.
>
> 9. **Lacking baselines with theoretical guarantees:** We thank the Reviewer for the constructive comments. In the paper, we discuss the proposed CFLAG, which provides theoretical guarantees for FCL with gradient aggregation. We also highlight the differences in the objectives of our proof compared to the proposed framework, demonstrating the significance of our theoretical contributions.

---

> ### Author Response · Authors · 2025-11-28
> **Following-up comment**
>
> Dear Reviewer CrLV,
>
> We would like to follow up on our earlier rebuttal. We understand the substantial workload during the review period and sincerely appreciate the time and effort you devote to evaluating the submissions.
>
> If your schedule allows, we would be grateful if you could also consider the additional elements included in our updated response, namely the revisions aimed at improving the paper’s presentation, the expanded discussion of the Theorems, and the additional theoretical analysis of gradient alignment. We would greatly value any feedback you may have on these points.
>
> Thank you once again for your time and consideration.
>
> Best regards,
>
> Authors

---

### Official Review · Reviewer_vU55 · 2025-11-01

**Soundness:** 2
**Presentation:** 2
**Contribution:** 2
**Rating:** 4
**Confidence:** 3

**Summary:**

This paper focuses on the federated continual learning (FCL) scenarios where heterogeneous tasks assigned to clients. The authors propose Spatio-Temporal grAdient alignMent with Prototypical coreset (STAMP), which uses gradient alignment method and prototypes for mitigating the biased feature learning and severe catastrophic forgetting.

**Strengths:**

* The paper focuses on an important research problem of CFL with heterogeneous clients.

**Weaknesses:**

* Gradient Alignment method is not something new
* The theoretical results are either trivial (Lemma 1) or too complex to get meaningful insight (Theorems 1,2). Especially, it was hard to find some connection between the theoretical results and the empirical observations.
* It is unclear when/why STAMP works well.

**Questions:**

None

---

> ### Author Response · Authors · 2025-11-24
> **Originality of gradient alignment method.**
>
> We acknowledge that gradient alignment is not a new concept, either in the broader optimization literature or within continual learning. Our contribution lies in demonstrating that gradient alignment provides an effective mechanism for jointly addressing both spatial heterogeneity and temporal catastrophic forgetting in heterogeneous FCL. Rather than treating spatial heterogeneity (i.e., clients encountering different tasks at the same time step) as a major obstacle, we leverage gradient alignment to identify an invariant gradient trajectory that generalizes across all participating tasks. This trajectory enables the global model to maintain consistent performance regardless of the local task distribution, thereby improving its task-level generalization.
>
> By enhancing generalization across tasks, the model inherently becomes more plastic, allowing it to more effectively acquire new knowledge without undermining previously learned information. Consequently, gradient alignment serves as a unifying principle that simultaneously improves generalization and mitigates forgetting in heterogeneous FCL. To enhance clarity, we have expanded the discussion of our contributions in the revised manuscript.

---

> ### Author Response · Authors · 2025-11-24
> **Clarification on the theoretical results**
>
> 1. **About the Lemma 1:** Lemma 1 is to decompose the distributional discrepancy between tasks from different clients into two separate distance terms. This decomposition is essential for enabling the disentanglement used in Theorem 1. The motivation for this disentanglement stems from the observation that existing FCL approaches primarily focus on mitigating catastrophic forgetting at the client level, while treating server-side aggregation similarly to standard FL. In contrast, our formulation leverages this disentanglement to develop a unified framework that jointly aggregates knowledge across both temporal and spatial dimensions. Following Reviewer’s suggestion, we have moved Lemma 1 to the Appendix to reserve space for additional substantive material to be included in the rebuttal.
> 2. **About the Theorem 1:** Theorem 1 establishes a relationship between the generalization gap of a continual learning model and that of an ideal model, expressed in terms of the distributional discrepancies between tasks across different time steps and across different clients. Unlike prior formulations of generalization bounds, our theorem explicitly decomposes this discrepancy into two components, temporal and spatial. This decomposition allows us to demonstrate that, rather than being an inherent limitation of heterogeneous FCL, the generalization gap can be further reduced by simultaneously mitigating both sources of discrepancy. To further clarify Theorem 1, we have added a detailed discussion in Section 3.1 of the revised manuscript.
> 3. **About the Theorem 2:** Similar to Theorem 1, Theorem 2 provides theoretical guarantees for heterogeneous FCL in terms of the generalization gap. This perspective allows us to characterize the model’s ability to generalize to previously encountered tasks (which reflects its level of catastrophic forgetting), as well as to unseen tasks (which reflects the model's plasticity). Theorem 2 shows that convergence is governed by five principal components: the bound associated with the ideal future task, the gradient divergence between tasks within a client, the gradient divergence between clients at a given time step, the statistical distance between observed tasks and future unseen tasks, and the gap induced by the finite sample size. Among these components, the first, fourth, and fifth terms cannot be reduced through algorithmic design. In contrast, the second and third terms can be mitigated through spatio and temporal gradient matching. Existing methods primarily focus on decreasing the temporal statistical discrepancy (second term) and consequently overlook its spatial counterpart (third term). Our approach explicitly targets the reduction of the second and third components, thereby narrowing the generalization gap between seen and unseen tasks. This leads to improved plasticity of the global model when deployed in heterogeneous FCL settings. In summary, this analysis highlights the key difference between our work and prior gradient alignment methods. A detailed explanation is provided in Sections 3.1 and 4 of the revised manuscript.

---

> ### Author Response · Authors · 2025-11-24
> **Discussion on the effectiveness of STAMP**
>
> 1. While Reviewer vU55 noted that Gradient Alignment is not novel, this also highlights its established effectiveness in improving model generalization. Consequently, the conditions under which STAMP performs well, and the reasons for its effectiveness, are well-supported.
> 2. We further elaborated on the theoretical underpinnings of STAMP in **Clarification on the theoretical results** comment, explaining the reasons for its effectiveness.
> 3. Additionally, we provided a detailed discussion in Appendix B in our revised manuscript on how the gradient alignment formula identifies invariant gradients.

---

> ### Author Response · Authors · 2025-11-28
> **Following-up comment**
>
> Dear Reviewer vU55,
>
> We would like to gently follow up on our previous rebuttal. We understand the substantial workload during the review period and sincerely appreciate the effort you dedicate to evaluating the submissions.
>
> If time permits, we would be grateful if you could also take into account the additional elements included in our updated response, i.e., the discussion on the Theorems and the additional theoretical analysis of gradient alignment. We would greatly value any feedback you may have on these points.
>
> Thank you once again for your time and consideration.
>
> Best regards, Authors

---

### Author Response · Authors · 2025-11-27

Dear Reviewers,

We are pleased to inform you that the additional experiments requested have now been completed. These experiments, which address the specific points raised in the reviews, have been incorporated into the revised version of the paper. For ease of reference, we have highlighted all additions and revisions in blue.

The new results further reinforce the claims made in the paper and provide additional clarity on the robustness and adaptability of STAMP in various settings. We encourage you to review these updates and welcome any further comments or suggestions you may have. Thank you once again for your time and consideration.

Besides, we also provided further theoretical results according to the gradient alignment and prototypical coreset to improve the paper clarity and contribution significance per the Reviewers' comments.

We also revised our paper to improve the representation, including fixing the notations, provided more detailed discussion to the Theorems, Algorithm to make the Prototypical Coreset Selection easier to be approached to the reader.

---

### Meta-Review · Area_Chair_UYSP · 2026-01-01

**Summary:**

This paper studies a challenging and practically relevant setting of federated continual learning with heterogeneous clients and sequential tasks. The authors propose STAMP, a spatio-temporal gradient alignment framework combined with a prototypical coreset, aiming to mitigate catastrophic forgetting while handling client and task heterogeneity. Reviewers generally agreed that the problem setting is important and that the paper is ambitious, combining algorithmic design, theoretical analysis, and empirical evaluation. Several reviewers also acknowledged the breadth of experiments and the attempt to move beyond generative replay-based approaches. However, substantial concerns were raised regarding novelty, clarity of contributions, theoretical depth, experimental interpretation, and whether the proposed approach convincingly demonstrates reduced catastrophic forgetting beyond existing methods.

**Reviewer Concerns:**

A central concern across reviews is that gradient alignment, which forms the core of STAMP, is not a new concept, and the paper does not sufficiently clarify what is fundamentally new beyond applying known alignment ideas to a federated continual learning setting. While the authors emphasize joint spatio-temporal alignment, reviewers found it unclear when and why this mechanism should work better than existing rehearsal- or alignment-based methods, and how it meaningfully differs from prior approaches in practice. The theoretical analysis was frequently described as either too weak or insufficiently connected to the empirical results: some reviewers viewed parts of the theory as trivial or descriptive rather than explanatory, while others noted that the bounds do not clearly capture the role of the prototypical coreset or directly justify reduced forgetting. On the empirical side, although additional experiments were provided in the rebuttal, reviewers remained concerned that the paper does not present sufficiently direct evidence of catastrophic forgetting mitigation, such as clear forgetting curves or task-level retention analyses, and that improvements in gradient alignment metrics do not necessarily translate to improved retention. Several experimental claims were also questioned, including potentially misleading interpretations of figures, unclear trade-offs between local and global accuracy, limited ablation coverage given the number of components introduced, and insufficient analysis of storage, scalability, and convergence costs in realistic large-scale settings. While the rebuttal addressed many points with additional experiments and clarifications, reviewers were divided on whether these additions fully resolved the conceptual and empirical gaps.

**Reviewer Scores:**

The reviewer scores reflect a lack of consensus and a wide spread of opinions. Multiple reviewers rated the paper marginally below the acceptance threshold or explicitly recommended rejection, often citing concerns about novelty, theoretical clarity, and the strength of evidence for forgetting mitigation. Several of these reviewers expressed moderate to high confidence in their assessments. At the same time, a smaller subset of reviewers viewed the work more favorably, rating it marginally above the acceptance threshold and highlighting the comprehensive experimentation, the use of prototypical coresets, and the attempt to unify spatial and temporal heterogeneity. Importantly, even reviewers with higher scores often explicitly stated that they would not mind rejection, indicating limited confidence that the contribution clearly meets the bar for ICLR. Overall, the score distribution and accompanying confidence levels suggest significant disagreement, with no strong consensus in favor of acceptance.

---

### Decision · Program_Chairs · 2026-01-26

Reject